# Routing in Solar-Powered UAV Delivery System

**Zijing Tian \*, Zygmunt J. Haas \* and Shatavari Shinde**

Computer Science Department, University of Texas at Dallas, Richardson, TX 75080, USA
* Correspondence: zxt140530@utdallas.edu (Z.T.); zjh130030@utdallas.edu (Z.J.H.)

**Abstract:** As interest grows in unmanned aerial vehicle (UAV) systems, UAVs have been proposed to take on increasingly more tasks that were previously assigned to humans. One such task is the delivery of goods within urban cities using UAVs, which would otherwise be delivered by terrestrial means. However, the limited endurance of UAVs due to limited onboard energy storage makes it challenging to practically employ UAV technology for deliveries across long routes. Furthermore, the relatively high cost of building UAV charging stations prevents the dense deployment of charging facilities. Solar-powered UAVs can ease this problem, as they do not require charging stations and can harvest solar power in the daytime. This paper introduces a solar-powered UAV goods delivery system to plan delivery missions with solar-powered UAVs (SPUs). In this study, when the SPUs run out of power, they charge themselves on landing places provided by customers instead of charging stations. Some advanced path planning algorithms are proposed to minimize the overall mission time in the statically charging efficiency environment. We further consider routing in the dynamically charging efficiency environment and propose some mission arrangement protocols to manage different missions in the system. The simulation results demonstrate that the algorithms proposed in our work perform significantly better than existing UAV path planning algorithms in solar-powered UAV systems.

**Keywords:** drones; path planning; solar-powered UAV; UAV-based delivery system

## 1. Introduction

As unmanned aerial vehicles (UAVs) become increasingly popular, their applications for drones/UAVs (drones and UAVs can be used interchangeably, although many professionals in the industry believe UAVs need to have autonomous flight capabilities, whereas drones do not; in this study, we only talk about UAVs in the form of quadcopters) are developed and expanded to many civilian or military areas. Soon, UAVs are predicted to play a significant role in the intellectualization of urban cities. However, the limited battery capacity makes it difficult for UAVs to finish long-distance/long-term missions without any power charging on the way.

In some UAV delivery systems, the solution is to build many charging stations along the way, so that UAVs can carry out longer missions. For example, in [1–3], charging stations were built for UAV power supply. Unfortunately, it is costly and sometimes illegal to build charging stations on rooftops or personal property in urban cities. Although some hosts are willing to have charging stations on their property, other issues arise, such as the power, the safety of the station and potential UAV-caused accidents. It is, therefore, challenging to build UAV charging infrastructures in both public and personal places.

Solar-powered UAVs can harvest sunlight power anywhere in the daytime given suitable weather. As a result, to achieve more extended distance/time missions by UAVs without building many charging infrastructures, we propose an autonomous solar-powered UAV delivery system to take over the traditional delivery work in urban cities. In our proposed system, places such as roofs of buildings (the study in [4] presented potential solar energy stations on rooftops in modern cities that can be easily discovered by UAVs), high grounds in parks and small places in people's gardens are identified as possible

charging places on city maps. It is much cheaper and more efficient to simply utilize such places as they are rather than building charging infrastructures there instead. In the proposed system, hundreds of solar-powered UAVs are prepared in stores with thousands of charging places determined throughout a city. When the system receives a delivery mission, it automatically finds the proper UAV that takes the least amount of time to finish the task. Since the UAVs can run out of power during the mission, the system helps with UAV path planning and determines how much time they need to stay on each passed landing place for solar power charging. In previous studies, authors paid more attention to path planning for solar-powered UAVs to harvest more energy during the flight (for fixed-wing UAVs), and no path planning methods were proposed for solar-powered UAV delivery systems. Thus, we believe it is necessary to comprehensively analyze the routing problem in solar-powered UAV delivery systems.

This study discusses the path planning problem by considering both the distance and the energy charging efficiency, since SPUs are applied here. The charging rates (efficiency) (the charging rate here represents the charging speed of a solar panel in a landing place in units of energy per minute, depending on the illumination/radiation level of the landing place) of different charging places differ in our scheme. Sequentially, we perform a trade-off between the distance and energy charging efficiency in this problem. The charging efficiency can either be static or dynamic in UAV systems. If the charging efficiency is considered static, it means the energy price in the normal UAV delivery system or the charging rate in the solar-powered UAV system is considered constant for each landing place. At the beginning of our study, both the optimal charging time assignment (CTA) algorithm and the globally optimal path planning algorithm (GOA) with pruning strategies are proposed to solve the path planning problem in the statically charging efficiency environment. As described in the solar radiation model proposed in [5], we also consider a dynamically charging efficiency in our study. It is much more efficient to charge the UAV in the period close to noon than in earlier or later daytime. Hence, the path planning problem became more complicated here. To solve the routing problem in the dynamically charging efficiency environment, we also propose a dynamically greedy CTA algorithm to further reduce the charging time on a known path. In addition, a heuristic path planning algorithm based on the dynamically greedy CTA algorithm is proposed to reduce the path finding time. Finally, some protocols are proposed to optimize the global UAV delivery works in the whole system.

The main contributions in this paper include the following:

1. We propose globally optimal CTA and path planning algorithms for UAV delivery problems in a statically charging efficiency environment. Previous studies have not proposed such globally optimal algorithms.
2. We propose a greedy CTA algorithm and a heuristic path planning algorithm for the solar-powered UAV delivery problem in a dynamically charging efficiency environment, which has never been proposed before.
3. We are the first to propose a SPU delivery system with a prototype, system design, and protocols to operate different missions.

The rest of this paper is organized as follows: Section 2 compares our study with related works. Section 3 proposes the path planning problems we encounter in the SPU delivery system. Section 4 proposes the CTA algorithm and globally optimal path planning algorithm for a single SPU mission in a statically charging efficiency environment. Section 5 proposes a dynamically greedy CTA algorithm and heuristic SPU path planning algorithm to solve the SPU routing problem in the dynamically charging efficiency environment. Section 6 proposes the mission arrangement protocols to solve the system-level problems. Section 7 proves the feasibility of applying SPUs in delivery works. Section 8 presents the simulation results for our proposed algorithms. The last section concludes the paper and suggests future work.

## 2. Related Studies

The subject of this paper was to solve the path planning problem in solar-powered UAV delivery systems, whereas no other studies explored the same topic as our study. Most papers about solar-powered UAV path planning have focused on fixed-wing solar-powered UAV path planning, which plans the optimal path to create wings with solar panels that can harvest the most power during the flight, such as [6,7]. However, since we aimed to work on the UAV delivery problems, it was not practical to apply fixed-wing solar-powered UAVs in this study. (Fixed-wing UAVs require runways to take off and land on, and it is not practical to build a lot of runways in modern cities. In addition, in our study, the optimization of charging time on a series of landing places was considered, whereas in fixed-wing solar-powered UAV path planning problems, the optimization of continuous charging efficiency on different flight paths is generally considered. Thus, we did not compare our proposed algorithms with any algorithms applied to fixed-wing UAVs.) As a result, we focused on the path planning problem for solar-powered quadcopters, which are more practical for delivery missions.

Some other studies discussed that the path planning problems of UAVs were related to obstacle avoidance, such as [8,9]. In our study, we ignored this kind of problem because we assumed that UAVs would fly to a safe altitude where no obstacles needed to be considered. Consequently, we focused on finding the best route for UAVs to finish a delivery mission with several landing times. Thus, we did not compare our algorithms with such kinds of studies.

Hence, we compared our study with papers related to quadcopter path planning problems and UAV delivery problems. As described in [10], since UAV path planning problems are always treated as being NP-hard (nondeterministic polynomial-time hardness) problems, the time complexity and complexity of the problems increase in complex terrains. They pointed out all the challenges for solar-powered UAV implementations, but did not give a particular solution to the UAV delivery path planning problem. In addition, different from fixed-wing UAVs, solar-powered quadcopter path planning problems are mainly affected by distance and solar power factors. Thus, in our study, we focused on the trade-off between energy charging and flying distance to find the best route. The authors of [11] concluded that in UAV routing problems, which require the finding of a set of locations as paths, authors are always trying to find a path that minimizes the operational cost. Accordingly, in our study, this cost stood in for the mission time, since no money was deducted during the solar-powered UAV delivery missions.

For UAV delivery research, such as [12–14], authors proposed routing algorithms for UAV delivery problems with the help of ground vehicles. Solar-powered UAVs are not appropriate for these delivery systems, because it takes more area to charge them than normal UAVs and requires a safe charging environment, while, in [15,16], the authors studied the path planning algorithm for the coverage plan of delivery customers. The UAV delivery system was discussed in some studies before, such as [17–19]. For example, in [17], the authors focused on the optimization of mission scheduling, while in our study, we focused on the optimization of path planning. Thus, the algorithms proposed in such kinds of studies cannot be compared with our study as well.

Studies from [1–3,20,21] were quite like our study. All these studies talked about path planning algorithms in UAV delivery systems with charging stations deployed. However, authors of [1,3,20] proposed heuristic algorithms to solve the path planning problems in UAV delivery systems, whereas authors of [2,10] proposed Dijkstra-based algorithms to solve the path planning problems. Since we are the first to propose path planning algorithms in solar-powered UAV delivery systems, it is quite difficult to compare our proposed algorithms with existing heuristic algorithms. The existing heuristic algorithms for UAV delivery path planning did not consider the factor of dynamically charging efficiency, which is very important in solar-powered UAV delivery routing problems. Additionally, in our study, we completed a comprehensive study of the optimization of solar-powered UAV routing problems, which has never been performed before. However,

it is difficult to compare our proposed algorithms with existing heuristic algorithms, since the weights of factors were different.

On the other hand, when compared to Dijkstra-based algorithms, our proposed algorithm was much better, because of the introduction of charging time assignment algorithms. In Dijkstra-based algorithms, authors assume that UAVs always charge themselves to a full battery state without considering CTA plans (in Dijkstra-based algorithms, the cost between nodes should be constant, and if UAVs are allowed to be partially charged in a charging station/landing place, the cost between nodes becomes a variable when the paths change). As a result, the Dijkstra-based algorithms cannot obtain globally optimal solutions in any scenarios where UAVs can be partially charged. More specifically, we proposed different CTA algorithms that modified the linear greedy algorithm proposed in [21] for vehicle recharging in the UAV delivery system. In a statically charging efficiency environment, our proposed CTA algorithm had an advantage compared with the linear greedy algorithm in SPU problems, because we could prune the path if any node was assigned 0 charging time by the linear greedy algorithm and reduce the total mission time in some cases. In a dynamically charging efficiency environment, our proposed dynamically greedy CTA algorithm further optimized the linear greedy algorithm with a local minimization operation and increased its efficiency when dealing with SPU path planning problems. With the great advantage brought by our proposed CTA algorithms for solar-powered UAV delivery systems, our proposed path planning algorithms showed obvious improvements compared to Dijkstra-based algorithms.

## 3. The Proposed Model of SPU Delivery System

### 3.1. SPU Delivery System

To replace the human work in delivery systems in urban cities, we proposed a system with solar-powered UAVs, which allowed for energy charging on landing places during their missions. There are two kinds of solar-powered UAVs, fixed-wing SPUs and multirotor SPUs. We chose multirotor solar-powered UAVs, because it is difficult for fixed-wing SPUs to lift and land in urban cities. In addition, it can be inefficient, and unsafe for multirotor SPUs to fly with their wings expanded during delivery missions when they are expected to carry heavy payloads. Thus, we preferred the transformable multirotor SPUs from [22], because, during the flight, the solar panels can be folded and unfolded again when landing to charge. The feasibility study of such an SPU is discussed in Section 7. Each UAV could only carry one package at a time, since the payload of a single UAV is limited.

Therefore, the solar-powered UAV delivery system could be built with three elements: delivery stores, solar-powered UAVs and landing places. The packages were stored at delivery stores, and these were the start nodes of the delivery missions. The solar-powered UAVs that carried out the delivery missions charged themselves on landing places. The landing places could either be intermediate nodes or the destination of the delivery missions, and in some emergency or the return voyages, they could be the start nodes as well.

In the proposed system, we assumed that the system would operate during the daytime, with optimal weather conditions for flight and solar power charging. As the energy required for flying was larger than the capacity of the battery, the UAVs periodically needed to land and stay grounded until having accumulated enough energy to continue their mission (this is reminiscent of the way that flying bugs travel, such as flies or bees, where these bugs periodically land to rejuvenate); we referred to such times as charging times. The charging rates at different locations were typically different and were assumed to be known (or approximately predictable) to the path planning facility. The goal of the proposed algorithms was to minimize the sum of total flying times and charging times, (see Equation (1) below). Thus, the proposed algorithm determined the path the UAV should follow, and how much time the solar-powered UAV should stay at each passed landing place. Note that for any UAV delivery mission, the path with the shortest distance may not be the path with the shortest total mission time. In other words, we determined a trade-off between flying distance and energy charging efficiency. In practice, the solar-powered

UAV path planning problem was more complicated, since our path planning algorithm was proposed for solar-powered UAVs and the charging rate in any landing place varied in the daytime; we also needed to find the best path planning algorithm in a dynamically charging efficiency environment. Finally, after finding the best path planning algorithm for each SPU delivery mission, the arrangement protocol for hundreds of missions was needed.

*3.2. SPU Routing Problem Definition (Statically Charging Efficiency)*

Before we optimized the whole SPU delivery system, we needed to solve the SPU routing problem for a single UAV. The ultimate objective of this path planning problem, called the single SPU problem, was to find a path that minimized the total mission time. The following diagram shows an example of a single SPU problem in our proposed system.

In Figure 1, the single SPU problem was designed to find a path from node 1 to node 10 with minimized mission time. Path 1-3-8-10 was a candidate optimal solution to this problem. The UAV started at node 1 then landed on nodes 3 and 8 to charge itself, and then arrived at node 10 if the system assigned this routing plan to it.

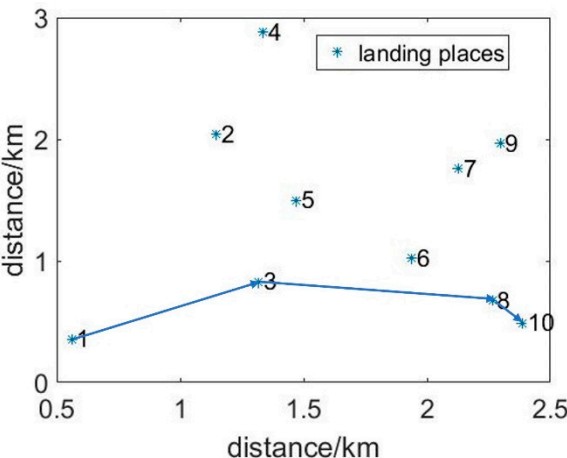

**Figure 1.** Example of a single SPU problem.

To simplify the routing problem, we ignored the lifting and landing time and energy costs in this routing problem, since they were negligible when we considered a long-term journey by the UAVs. However, it was still not simply a problem of finding the shortest path, since there was a trade-off between distance and energy charging for any path. For example, even if the length of a path was short, the total mission time would still be long because the charging efficiencies of the nodes along this path were low, and it would take much more time for the UAV to charge along this path. To deal with this issue, we simplified the path planning problem with Assumptions 3.2.1–3.2.3, discussed below.

The path planning problem was equivalent to a mission time minimization problem:

$$minT_{total} = min\left(T_{total-flight} + T_{total-charge}\right) \tag{1}$$

where $T_{total-flight}$ represents the total flying time and $T_{total-charge}$ represents the total charging time during the mission. $T_{total}$ is the total mission time that we wanted to minimize. Further, we assumed that the charging efficiency at node x, defined as the fraction of the maximum charging rate of the UAV's energy storage, was $\rho_x$. If node x had the maximal possible charging rate (i.e., sufficiently large radiation that allowed for the maximal charging rate of the UAV energy storage), $\rho_x = 1$, then, in general, $0 < \rho_x \le 1$.

We defined the problem formally as follows: given a geographical (local) map with marked n locations (called nodes): $V_{local} = \{v_1, v_2, \ldots, v_n\}$, where $v_1 = v_S$ is the start node and $v_n = v_D$ is the destination, and $v_2, \ldots, v_{n-1}$ are possible landing places on the geographical map. At first, a graph was generated to describe the reachability relation between the nodes $G = (V_{local}, E_{reachable})$. $E_{reachable} = \{e_1, e_2, \ldots, e_m\}$ represents all the

edges that follow $l_{e_i} \leq d_{max}$, where $l_{e_i}$ is the length of edge $e_i$, and $d_{max}$ is the maximum one nonstop flying distance of the UAV with its energy storage maximally charged. Then, the SPU path planning problem could be described as the mission time minimization problem with Equation (1) when graph G and distance $d_{max}$ were chosen.

Let $l_{S-D}$ denote the straight-line cross node $v_S$ and node $v_D$. Additionally, let $e_{ij}$ denote the edge between nodes $v_i$ and $v_j$.

**Assumption 3.2.1.** *G is a graph as defined above for all the edges in $E_{reachable}$. Line $l_{S-D}$ needs to be the parallel line of the X-coordinate in the local city map. The coordinate of node $v_S$ is $(0, Y_S)$ and the coordinate of node $v_D$ is $(X_D, Y_S)$. For any edge in $E_{reachable}$ as $e_{ij}$, assume the coordinate of node $v_i$ is $(X_i, Y_i)$ and the coordinate of node $v_j$ is $(X_j, Y_j)$, always keeping $X_j \geq X_i$.*

This assumption ensured that the UAV did not fly significantly "back" to nodes away from the destination. Additionally, circulations in paths were also eliminated with this assumption.

For any given path $P_k = \left[ v_S, v_{k_1}, v_{k_2} \dots v_{k_j}, v_D \right]$, where $v_{k_1}, \dots, v_{k_j}$ were possible landing places along the path, $T_{total-flight}$ was fixed, because the speed of the UAV and the distance between any two nodes were fixed in G. As a result, Equation (1) could be written as:

$$\min T_{total} = T_{total-flight} + \min\left( T_{total-charge} \right) \tag{2}$$

Assuming that the number of nodes in $P_k$ is L, $T_{total-flight} = \sum_{q=0}^{L-2} T_{flight(max)} \times \frac{d_{k_q \to k_{q+1}}}{d_{max}}$, where $T_{flight(max)}$ is the maximum flying time for a fully charged UAV, $d_{max}$ is the maximum distance a UAV could fly with a fully charged energy storage, $d_{k_q \to k_{q+1}}$ represents the distance between node $v_{k_q}$ and node $v_{k_{q+1}}$ ($v_{k_q}$ is the $(q+1)^{th}$ node in $P_k$), and the start node $v_S \equiv v_{k_0}$ on path $P_k$. $T_{total-charge} = \sum_{i=1}^{L-2} T_{charge(k_i)}$, where $T_{charge(k_i)}$ is the charging time in node $k_i$. Subsequently, if the path is known as $P_k$, the total mission time along path $P_k$ can be calculated as:

$$T_{total} = \sum_{q=0}^{L-2} T_{flight(max)} \times \frac{d_{k_q \to k_{q+1}}}{d_{max}} + \sum_{i=1}^{L-2} \frac{T_{charge-est(k_i)}}{\rho_{k_i}} \tag{3}$$

where $T_{charge-est(k_i)}$ represents the estimated charging time at node $k_I$ when $\rho_{k_i} = 1$. Then, the actual charging time for any node $k_I$ could be obtained with $T_{charge(k_i)} = \frac{T_{charge-est(k_i)}}{\rho_{k_i}}$. $T_{total-charge}$ depended on the charging time assignment plans along a path $P_k$; e.g., one possible scheme was that we assumed that the UAV was charged only to the level needed to reach the next node on the path. In general, the approximate range of $T_{total-charge}$ could be found with the following formula:

$$T_{charge(max)}^{\rho_{max}} \times \frac{T_{total-flight} - T_{flight(max)}}{T_{flight(max)}} \leq T_{total-charge} \leq T_{charge(max)}^{\rho_{min}} \times \frac{T_{total-flight} - T_{flight(max)}}{T_{flight(max)}} \tag{4}$$

where $\rho_{min}$ represents the minimum charging efficiency for all charging places on a local city map, $\rho_{max} = 1$ represents the maximum charging efficiency for all charging places and $T_{charge(max)}^{\rho}$ represents the total time to charge the battery from empty to a fully charged state when the charging efficiency was $\rho$. $T_{charge(max)}^{\rho_{max}} = \rho_{min} T_{charge(max)}^{\rho_{min}}$.

Assuming that $E_p$ was the remaining power in the UAV when the UAV reached the $(p+1)^{th}$ node on path $P_k$, $R_p$ represented the charging time to charge the UAV battery to $E_p$ when charging efficiency was $\rho_{max}$. Assumption 3.2.2 was created to simplify the derivation:

**Assumption 3.2.2.** $R_1 = R_S = T_{charge(max)}^{\rho_{max}}$; *if* $T_{total-flight} > T_{flight(max)}$, $R_L = R_D = 0$.

Assumption 3.2.2 supposes that we always had a fully charged UAV at the beginning of a mission. If the UAV could not fly from node S to D without recharging, meaning that $T_{total-flight} > T_{flight(max)}$, we assumed that when the UAV reached the destination it had no remaining power. On the other hand, if $T_{total-flight} \leq T_{flight(max)}$, we did not need to consider the energy charging time of any path, because we did not need to charge the UAV during its mission.

We also ignored the landing and lifting-off times, as those are typically negligible relative to the flight time between two nodes on a path:

**Assumption 3.2.3.** *The lifting-off times and the landing times of each mission were assumed to be negligible.*

Besides the assumptions above, we needed to claim that, in this study, the energy charging time and distance between nodes were the only two factors we took into consideration. Here, we assumed a fixed flying speed for all solar-powered UAVs in this problem to simplify the path planning problem, since energy and distance are more crucial for solar-powered path planning problems.

In addition, to finding an optimal charging time assignment plan for any path $P_k$, we recorded the sets of reachable nodes for every node $v_I$ in $V_{local}$ as $U_i = \{U_1, U_2, \ldots, U_n\}$. Any node $v_j$ in reachable set $U_I$ should follow $d_{i \to j} \leq d_{max}$ and $X_j \geq X_i$. For example, in Figure 2, for the start node $v_s$, which was node one, nodes two, three and four were in the blue circle with the center as node one and the radius as $d_{max}$. Consequently, the reachable set of node one was $U_1 = \{2, 3, 5\}$.

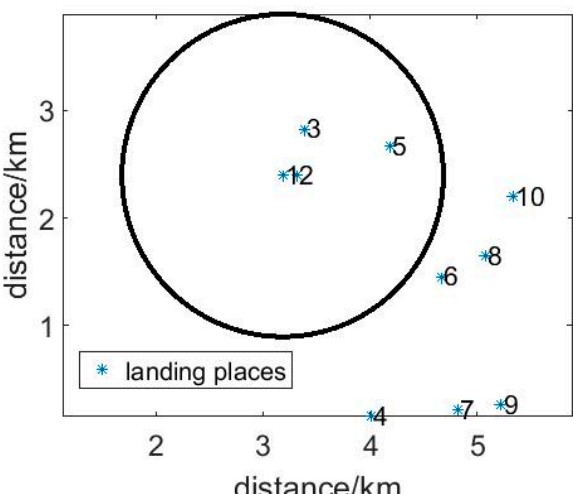

**Figure 2.** Reachable nodes of a node in the local city map.

The routing solution in the statically charging efficiency environment, as described by Equations (1) and (3), was solved in Section 4.

### 3.3. SPU Routing Problem Definition (Dynamically Charging Efficiency)

In our SPU path planning problem discussed above, we considered the charging efficiency $\rho_{k_i}$ of the node $k_i$ as a fixed value. However, in practice, the charging efficiency is a variable and varies during the daytime; thus, it is time dependent: $\rho_{k_i}(t)$. Solar radiation/illumination prediction is of great importance for many applications in a modern city [5], and we could see from [5] that the solar radiation in a place can be approximately represented by a curve.

In our study, we adopted the time-dependent solar radiation model called the S. Kaplanis's model from [5], shown in Figure 3 (the *Y*-axis label was replaced with charging efficiency, as defined in this paper):

$$\rho_{k_i}(t) = \left(a_{1(k_i)} + a_{2(k_i)} \cos \frac{\pi\left(t - t_{peak}\right)}{t_{end}}\right) \tag{5}$$

where $t_0 = 0$ represents the time of 6:00 a.m., $t_{peak} = 360$ is the peak time of radiation, which was noon, and $t_{end} = 720$ represents the end of the daytime, which was 6:00 p.m. (since the unit of time in our study was minutes). Taking the example from [23], where the mean annual solar radiation in the state of Hawaii was shown, we could say that in different places in the same city, radiations are different, so the prediction factors $a_1$ and $a_2$ in Equation (5) were location dependent.

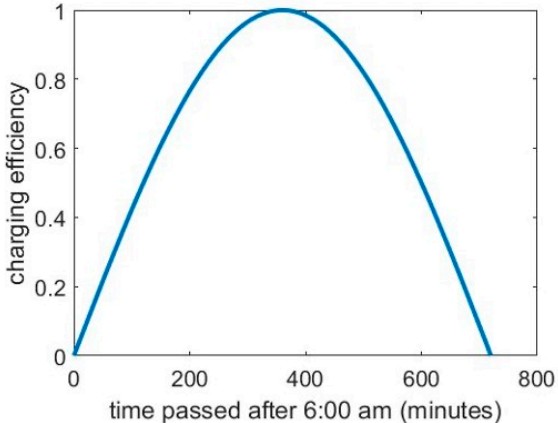

**Figure 3.** S. Kaplanis's charging efficiency prediction from [5].

Thus, based on the radiation data collected in any potential landing place, we could generate the radiation prediction formula per Equation (5) with a simulation of the detrending model in [5]. Then, the total mission time along path $P_k$ could be expressed with Equation (6):

$$T_{total} = \sum_{q=0}^{L-2} T_{flight(max)} \times \frac{d_{k_q \to k_{q+1}}}{d_{max}} + \sum_{i=1}^{L-2} \frac{T_{charge-est(k_i)}\left(t_{i1} - t_{i0}\right)}{\int_{t_{i0}}^{t_{i1}} \rho_{k_i(t)} \cdot dt} \tag{6}$$

where $t_{i0}$ represents the time that the UAV arrived at node $k_i$ and $t_{i1}$ represents the time the UAV left node $k_i$ at. When $t_{i0}$ and $t_{i1}$ for each node were determined, we could find the value of $T_{total}$ with Equation (6). To solve the total mission time minimization problem described by Equation (6), we studied how these variables affected the total mission time and tried to find the corresponding algorithm for the dynamically charging efficiency environment in Section 5.

### 3.4. Mission Arrangement of Problems Definition

The final problem of our study was to plan a UAV delivery system with multiple UAVs operating at the same time. There could be a large number of possible landing places on a city map. We assumed to have a multitude of C missions $M = \{m_1, m_2, \ldots, m_C\}$ to perform every day, and there was a set of UAVs (with B UAVs prepared) $UAV\{UAV_1, UAV_2, \ldots, UAV_B\}$ that could be used to execute these missions.

Considering the coordination of the missions in the whole city, the problem described by Equation (1) was changed to:

$$\min T_{total-mission} = \sum_{i=1}^{c} T_{total_i} \qquad (7)$$

where $T_{total_i}$ is the total time of the $i^{th}$ mission, all the missions had to be completed and the total mission time $T_{total-mission}$ had to be minimized.

In Figure 4, the ten stores were marked with red circles and one hundred possible landing places were marked with blue crosses. The mission destinations were some of the landing places.

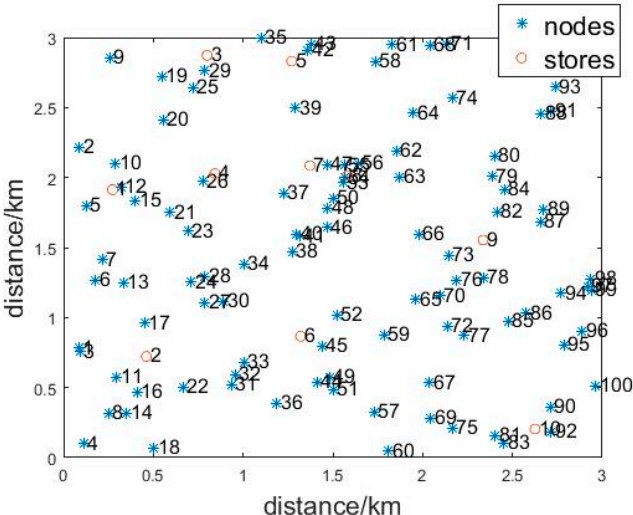

**Figure 4.** City map with landing places (nodes) and stores.

In the whole proposed SPU delivery system, we needed to consider the following problems:

1. Whether the space in each landing place was limited.
2. Whether the local city map needed to be updated from time to time.
3. Whether the system should be able to handle emergencies caused by bad weather.

Section 6 further addresses the above problems by our proposed protocols.

## 4. Proposed Routing Algorithms (Statically Charging Efficiency)

In Section 2, we mentioned that the algorithm in [2,3] did not yield a globally optimal solution. Their CTA plan was not optimal for any given path because, in their assumption, the UAVs always charged their battery to a full battery state. For example, in Figure 5, on the 1-5-9-10 path, let us say that the charging efficiency on node eight was much larger than on node three. When we reached node three, as opposed to fully charging the battery, we only needed to charge enough to reach node eight. This plan was more efficient than plans determined by [2,3]. When we considered both the total flying distance and energy charging efficiency in the path planning problem, even though the path found by the Dijkstra algorithm had a minimized distance (the distance for the Dijkstra algorithm here was from a drone navigation and charging station (DNCS) [2], which was calculated with the summation of flying time and charging time under the scenario of UAVs always charging themselves to a full battery capacity in any station/landing place [3]) of a mission, the path found by this Dijkstra algorithm may not have been the optimal one. For example, in Figure 5, assume that the Dijkstra algorithm found an optimal path 1-5-9-10 for the following graph. However, node nine had a much higher charging efficiency than nodes along path 1-5-9-10. Thus, if we charged the UAV on node nine on path 1-4-5-9-10, the total mission time $T_{total}$ could be smaller than that on path 1-5-9-10 obtained with Dijkstra's

algorithm. Thus, to find a globally optimal path planning solution for the SPU delivery mission, we needed a principle to traverse all the possible paths and to find the optimal charging time assignment along each such path.

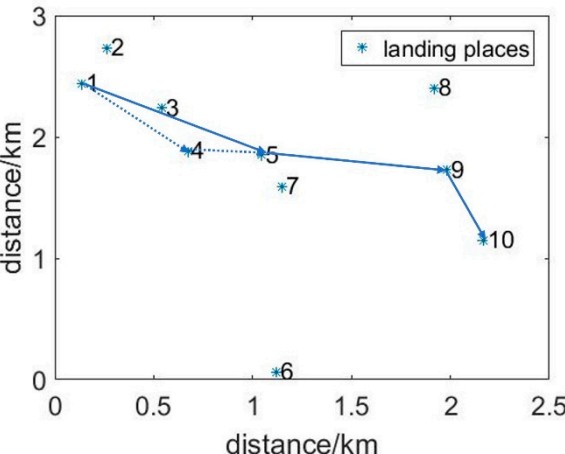

**Figure 5.** An example of a ten-node local city map.

The algorithm proposed in this section found a globally optimal solution of the SPU path planning problem in a statically charging efficiency environment with an acceptable running time complexity (our proposed algorithms worked on local city maps with at most hundreds of nodes).

*4.1. Changing Time Assignment Algorithm*

The algorithm presented in this section was a simple globally optimal algorithm (GOA) for solving the problem described in Section 3.2. Namely, the solution to this problem was just a combination of the path traverse algorithm with a depth first search (DFS) and the charging time assignment (CTA) algorithm. The CTA algorithm was necessary in the case that a UAV was unable to reach the destination and needed to find the next landing place for charging before completing the mission.

If a UAV could not fly without recharging to the destination from the current node (including the start node), it needed to find the next landing place for energy charging. This was where the CTA algorithm was necessary.

Assuming the UAV at the time was in the $w^{th}$ node on path $P_k$ ($w \geq 1$), it could reach the next h nodes along path $P_k$ (which meant the distances between the $w^{th}$ node and its next h nodes were smaller than $d_{max}$) and the flight time to reach the current $w^{th}$ node was $T_{flight-total(w)}$. For any $w^{th}$ node on path $P_k$, assume that the reachable set for the $w^{th}$ node was $U_{P_k(w)}$, and the set $u_{P_k(w)}$ ($u_{P_k(w)} = P_k \cap U_{P_k(w)}$); i.e., $u_{P_k(w)}$ contained only the nodes that were both in $P_k$ and $U_{P_k(w)}$. Let set $Z_{P_k(w)} = \rho_{k_{(w-1)}}$ denote the charging efficiency of the $w^{th}$ node on path $P_k$ and $P_{X-Y}$ denote the part of the path from the $X^{th}$ node to the $Y^{th}$ node.

**Theorem 4.1.** *If there exists o ($o > 1$) such that the $(w + o)^{th}$ node on the path $P_k$ has a larger charging efficiency than all the other nodes in $u_{P_k(w)}$, path $P_k$ cannot be an optimal path.*

**Proof of Theorem 4.1:** Assume the $(w + o)^{th}$ node ($o > 1$) has a larger charging efficiency than all the other nodes in $u_{P_k(w)}$. Additionally, assume two charging plans A and B. Plan A included all the nodes on the path in $P_k$, and in plan B, we skipped the nodes between the $w^{th}$ node and the $(w + o)^{th}$ node. Then, $T_{total-flight(planB)} \leq T_{total-flight(planA)}$, because the distance of the straight line between the $w^{th}$ and $(w + o)^{th}$ nodes in $P_k$ was never larger than any other path between the nodes that went through other nodes. In fact, $T_{total-charge(planB)} < T_{total-charge(planA)}$, because we did not waste time to charge the UAV

on nodes between the $w^{th}$ node and the $(w+o)^{th}$ node, of which the charging efficiencies were lower than the $(w+o)^{th}$ node. Thus, $T_{total(planB)} < T_{total(planA)}$. Accordingly, plan A with path $P_k$ could not be an optimal path. $\square$

**Lemma 4.1.** *Assume we are currently in $w^{th}(w>1)$ node in $P_k$. If any Y node before the $w^{th}$ node along the path $P_k$ can reach the $w^{th}$ node, and node Y has a larger charging efficiency than any other node along the part of $P_k$ from w to Y, $P_k$ should be pruned.*

**Remark 4.1.** *With the linear greedy algorithm in [21], if we pruned all the paths that satisfied Lemma 4.1, then we would have the following equation to calculate the total charging time on the $(w+1)^{th}$ node in $P_k$:*

$$\text{If } \max\{Z(w+1)\} < \rho_{k_{(w)}}, T_{charge-total(w+1)} = T_{charge-total(w)} + \frac{T^{\rho_{max}}_{charge(max)}}{\rho_{k_{(w)}}}$$

$$\text{If } \max\{Z(w+1)\} \geq \rho_{k_{(w-1)}} \text{ or } w = L-2, \; T_{charge-total(w+1)}$$
$$= T_{charge-total(w)} + \frac{\left(T^{\rho_{max}}_{charge(max)} \times \frac{d_{k_w \to k_{w+1}}}{d_{max}} - R_w\right)}{\rho_{k_{(w)}}} \qquad (8)$$

where $\{Z(w+1)\}$ represents the set of charging efficiencies of nodes in $u_{P_k(w+1)}$. If we charged the UAV to full capacity in the $w^{th}$ node, $R_{w+1} = T^{\rho_{max}}_{charge(max)} - T^{\rho_{max}}_{charge(max)} \times \frac{d_{k_{w-1} \to k_w}}{d_{max}}$; otherwise, $R_{w+1} = 0$.

It was shown in [21] that if there existed an optimal solution, the greedy algorithm of Equation (8) could discover it.

We, hereby, present Algorithm 1 to find an optimal CTA on path $P_k$:

---

**Algorithm 1:** CTA algorithm.

---

Input: Path $P_k$, graph G, charging efficiency Z for all the nodes in $V_{local}$, reachable set U
Output: Minimized total charging time $T_{charge-total}$ along $P_k$ and charging time on each node $k_i$, which is $T_{charge(k_i)}$
Variables: w, v
1. For $w = 1 : L-1$
2.      For $v = w-1: 1$
3.          If the $(w+1)^{th}$ node is in $u_{P_k(w)}$ and $Z_{P_k(w+1)}$ is larger than any other node along $P_{v-(w+1)}$, path $P_k$ is dropped (by Lemma 4.1)
4.              Break
5.          End//drop nonoptimal paths
6.      End
7.      If $P_k$ is kept, the next node is the next landing place. Additionally, in this $(w+1)^{th}$ node, the UAV is charged and allowed to have enough power to reach the $(w+2)^{th}$ node. Update $T_{charge-total(w+1)}$ with Equation (8)
8.      End//assign charging time according to Equation (8)
9. End
10. Return $T_{charge-total(L)}$ and $T_{charge(k_i)}$

---

Algorithm 1 was applied to find the optimal charging time assignment plan in any statically charging efficiency environment and could be tested by comparing the total charging time it found with what the other algorithms found. Since the CTA algorithm took $O(n^2)$ and time path generation algorithm took $O(n^2)$, the total running time was $O(n^4)$ if we just processed the CTA algorithm on all the possible paths and then found the optimal path.

Reference [21] proposed a related algorithm for vehicle refueling policies. In their study, cars needed to find a reachable refueling place with the lowest price at each stop. However, in UAV path planning problems, if we wanted to skip a node along a path, it

would not be necessary to pass this node, and we could just fly straight to the next node. As a result, in our SPU path planning problem, the path was changed when we decided to skip a node along the path. In addition, our proposed Algorithm 1 was different from the linear greedy algorithm in [21]. For example, assume we had two nodes A and B on path P, and there were some nodes between A and B on path P. In the scenario from [21], to decide if A could reach B, the total distance along A to B on path P should be smaller than $d_{max}$. However, in our SPU routing problems, A reaching B meant that the cartesian distance between A and B was smaller than $d_{max}$.

### 4.2. Globally Optimal Routing Algorithm

As discussed above, nonoptimal paths were dropped during the CTA algorithm. However, it was more efficient to prune these nonoptimal paths during the path generation process instead of the CTA process. As Lemma 4.1 claims, in each branch generated with the DFS algorithm, for each new generated node $v_i$ along path P, if any node along path P before $v_i$ could reach $v_i$ and all the nodes between them had lower charging efficiencies than $v_i$, path P should be pruned. Therefore, we skipped node $v_i$ and found the next node to generate a new path. If there was no path allowing the UAV to finish the delivery mission, the system reported a failure.

Another pruning strategy was to use the result of the Dijkstra algorithm as a guide to evaluate if a generated path could be a candidate for a globally optimal path. Combining this with the strategy based on the CTA, the complete GOA with a pruning strategy for SPU is depicted in Figure 6.

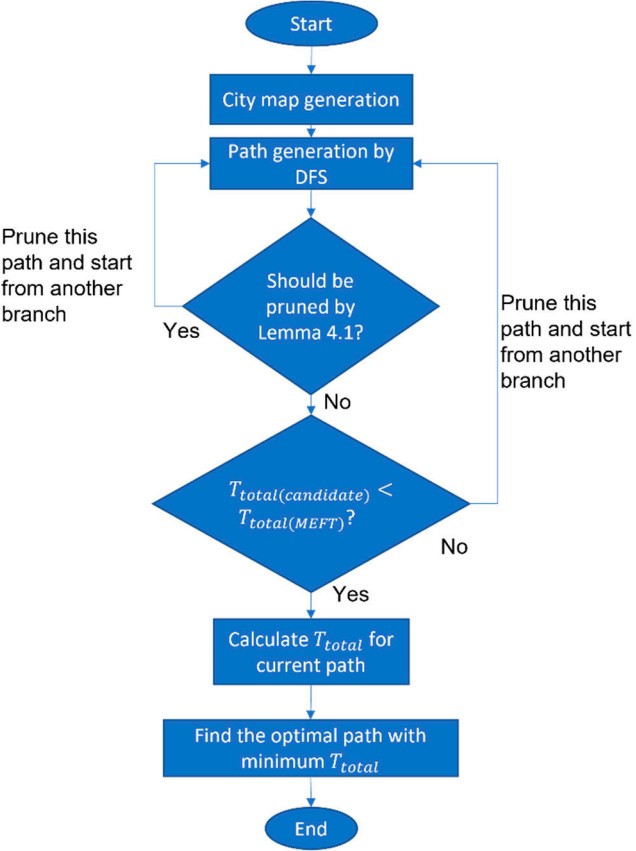

**Figure 6.** The GOA algorithm with pruning strategy.

The result of the Dijkstra algorithm represented an upper bound for the total mission time, and at the beginning of the GOA algorithm processing, the result of the Dijkstra algorithm was set as the maximal estimated flying time (MEFT), $T_{total(MEFT)}$:$T_{total(MEFT)} =$

$T_{total(dijkstra)}$. With this bound in mind, we calculated the total mission time for the current path, $T_{total(candidate)}$, with the iterative application of Remark 4.1 and general Equation (3).

Algorithm 2 shows the detailed process for GOA with pruning.

---

**Algorithm 2:** GOA with pruning.

---

Input: Graph G, $T_{total(MEFT)} = T_{total(dijkstra)}$, charging efficiency Z for all nodes in $V_{local}$, reachable set U

Output: Minimized total mission time $T_{total}$ and the optimal path $P_{optimal}$

Variables: L, w, v

1. While all the possible unpruned paths are generated,
2.      Process the DFS path to find the algorithm and initialize the current generated path to $P_k = \left[ v_S, v_{k_1}, v_{k_2} \ldots v_{k_{L-1}} \right]$, and assume to now be in the $w^{th}$ node in $P_k$ which is $v_{k_j}$ and the next node of $P_k$ is $v_{k_{j+1}}$. Additionally, the total mission time arriving node $v_{k_j}$ is $T_{total(candidate)}$.//Path generation
3.      For v = w −1: 1,
4.          If the $(w + 1)^{th}$ node is in u $_{P_k(w)}$ and Z $_{P_k(w+1)}$ is larger than any other node along $P_{v-(w+1)}$, path $P_k$ is pruned and any path starting with $P_k$ is pruned as well
5.             Break//pruning strategy by Lemma 4.1
6.          End
7.      End
8.      If $P_k$ is unpruned by Lemma 4.1, add $v_{k_{j+1}}$ to $P_k$ and calculate $T_{total(candidate)}$ when $v_{k_L}$ is added
9.          If $T_{total(candidate)} > T_{total(MEFT)}$,
10.             $P_k$ is pruned and any path starting with $P_k$ is pruned as well
11.          Else, if $T_{total(candidate)} \leq T_{total(MEFT)}$ and $v_{k_L} = v_D$
12.             $T_{total(MEFT)} = T_{total(candidate)}$ and $P_{optimal} = P_k$
13.          Else, if $T_{total(candidate)} \leq T_{total(MEFT)}$ and $v_{k_L} \neq v_D$
14.             Back to step 1
15.          End//find candidate optimal path by comparing
16.      End
17. Return $T_{total(MEFT)}$ and $P_{optimal}$

---

Algorithm 2 was applied to find the optimal path for the SPU delivery problem in the statically charging efficiency environment and could be tested by comparing the total mission time it found with what other algorithms found. The running time of the unpruned GOA was $O(n^4)$. However, since we only processed the CTA on unpruned paths, we only needed to calculate the total mission time with Equation (8) on these paths. Consequently, the running time of the pruned GOA could save time exponentially, according to the experiment in Section 8.1. Additionally, the great improvement by the GOA in the experiment also showed that it was necessary to carry this out, because we could significantly save customer time with the GOA. (The GOA could also be applied in other kinds of UAV path planning problems, where the charging efficiency is static. For example, in the path planning problem from [2], when the cost coefficient of time and money was fixed, the charging (cost) efficiency became a fixed value, which was related to the price of energy in different nodes. In such kinds of problems, the GOA could be applied and an optimal solution could be obtained as well.)

## 5. Proposed Routing Algorithms (Dynamically Charging Efficiency)

As described in Section 3.3, the charging efficiency for any landing place varied with time throughout the day, as shown by Equation (5). Given path $P_k = \left[ v_S, v_{k_1}, v_{k_2} \ldots v_{k_j}, v_D \right]$, the estimated total mission time could be calculated with Equation (6). However, there were different kinds of variables in Equation (6), such as $T_{charge-est(k_I)}$, $t_{i0}$ and $t_{i1}$, which caused it to be difficult to find a minimized solution of the total mission time. Thus, in this section,

we further analyzed this minimization problem and proposed SPU path planning algorithms regarding the dynamically charging efficiency.

## 5.1. Advanced Routing Problem Analysis (Dynamical Charging Efficiency)

In any known path $P_k = \left[ v_S, v_{k_1}, v_{k_2} \ldots v_{k_j}, v_D \right]$ in graph $G = (V_{local}, E_{reachable})$, represented in Figure 7, the total mission time was calculated in Equation (6), when the charging efficiency was dynamic.

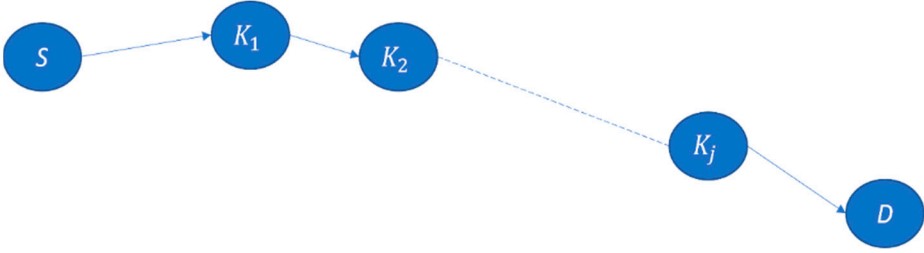

**Figure 7.** Path $P_k$, which contains nodes $\left[ v_S, v_{k_1}, v_{k_2} \ldots v_{k_j}, v_D \right]$.

Since, in Equation (5), we could not compare the real-time charging efficiency of two different landing places when we did not know the time t the UAV would arrive at these charging places, the pruning strategy by Lemma 4.1 in Section 4.1 could not be applied in the SPU path planning problem with dynamically charging efficiency. It was also difficult to compare the total mission time of two paths unless we found a proper charging time assignment algorithm for each path, as well as there being too many variables in Equation (6). Please note that, in Equation (6), the total flying time for a UAV mission, which was $T_{total-flight} = \sum_{q=0}^{L-2} T_{flight(max)} \times \frac{d_{k_q \to k_{q+1}}}{d_{max}}$, was only determined with path $P_k$; the total mission time minimization problem, therefore, was only determined by the total charging time, which was $T_{total-charge} = \sum_{i=1}^{L-2} \frac{T_{charge-est(k_i)}(t_{i1} - t_{i0})}{\int_{t_{i0}}^{t_{i1}} \rho_{k_i(t)} \cdot dt}$. Hence, if we found the minimized total charging time with a CTA plan and then compared the total mission time for all possible paths from node S to node D, the mission time minimization problem could be solved.

Firstly, Equation (6) needed to be further analyzed to find a more general description of the total mission time minimization problem. Thus, in this section, we tried to simplify the variables in Equation (6). We started by studying the relationship between $T_{charge-est(k_i)}$ and $t_{i0}$, $t_{i1}$. In any intermediate node (nodes except for the start node and destination) $k_i$ on path $P_k$, the exact charging time $T_{charge(k_i)}$ could be calculated with Equation (9):

$$T_{charge(k_i)} = \frac{T_{charge-est(k_i)}(t_{i1} - t_{i0})}{\int_{t_{i0}}^{t_{i1}} \rho_{k_i(t)} d_t} \tag{9}$$

where $\frac{\int_{t_{i0}}^{t_{i1}} \rho_{k_i(t)} d_t}{(t_{i1} - t_{i0})}$ represents the averaged charging efficiency on node $k_i$ between times $t_{i0}$ and $t_{i1}$. Moreover, $T_{charge(k_i)}$ could also be calculated with $T_{charge(k_i)} = t_{i1} - t_{i0}$. Thus, we had the following equation:

$$\frac{T_{charge-est(k_i)}(t_{i1} - t_{i0})}{\int_{t_{i0}}^{t_{i1}} \rho_{k_i(t)} d_t} = t_{i1} - t_{i0}$$

$$\int_{t_{i0}}^{t_{i1}} \rho_{k_i(t)} d_t = T_{charge-est(k_i)} \tag{10}$$

If we wanted to charge the UAV from an empty to full battery state, $T_{charge-est(k_i)} = T^{\rho_{max}}_{charge(max)}$. (In addition, with dynamically charging efficiency, we always had $T_{charge(k_i)} > T_{charge-est(k_I)}$, since $\rho_{k_i(t)} \leq 1$ and $\frac{\int_{t_{i0}}^{t_{i1}} \rho_{k_i(t)} d_t}{t_{i1} - t_{i0}} < 1$).

With Equation (5) and Figure 3, we could find the relations between $a_{1(k_i)}$ and $a_{2(k_i)}$:

(1)  $a_{1(k_i)} = 0$; with Figure 3, when t=$t_0$ = 0, we had $\rho_{k_i(t)} = 0$. Thus, $(a_{1(k_i)} + a_{2(k_i)} \cos \frac{\pi(t-360)}{720}) = a_{1(k_i)} = \rho_{k_i(t)} = 0$;

(2)  Assuming the peak charging efficiency for node $k_i$ was $\rho_{k_i (peak)}$, we always had $a_{2(k_i)} = \rho_{k_i (peak)}$. With Figure 3, when t = $t_{peak}$ = 360, $\rho_{k_i(t)} = \rho_{k_i (peak)}$. Thus, $a_{2(k_i)} \cos \frac{\pi(t_{peak}-360)}{720}) = \rho_{k_i (peak)}$, and then we had $a_{2(k_i)} = \rho_{k_i (peak)}$. $\rho_{k_i (peak)}$ could be estimated with the used data collected in different landing places.

Thus, Equation (10) was changed to Equation (11):

$$\int_{t_{i0}}^{t_{i1}} \rho_{k_i (peak)} \cos \frac{\pi(t-360)}{720} d_t = T_{charge-est(k_i)}$$

$$\rho_{k_i (peak)} \frac{720}{\pi} \sin \frac{\pi(t_{i1}-360)}{720} - \rho_{k_i (peak)} \frac{720}{\pi} \sin \frac{\pi(t_{i0}-360)}{720} = T_{charge-est(k_i)} \quad (11)$$

If $T_{charge-est(k_i)}$ and $t_{i0}$ were known, we could solve the equation above with Equation (12):

$$t_{i1} = \frac{720}{\pi} \sin^{-1}(\frac{\pi}{720\rho_{k_i (peak)}}(T_{charge-est(k_i)} + \rho_{k_i (peak)} \frac{720}{\pi} \sin \frac{\pi(t_{i0}-360)}{720})) + 360 \quad (12)$$

where $t_{i0}$ could be obtained with Equation (13):

$$t_{i0} = t_{(i-1)1} + T_{flight(max)} \times \frac{d_{(k_{i-1}) \to (k_i)}}{d_{max}} \quad (13)$$

where $t_{(i-1)1}$ represents the time the UAV left node $k_{i-1}$ and $T_{flight(max)} \times \frac{d_{(k_{i-1}) \to (k_i)}}{d_{max}}$ represents the total flying time from node $k_{i-1}$ to node $k_i$. It is obvious that Equation (13) is an iterative equation, where each $t_{i0}$ and $t_{i1}$ in node $k_i$ are determined not only by the time the UAV leaves the previous node $k_{i-1}$, but also by the estimated charging time $T_{charge-est(k_i)}$ for node $k_i$. However, if the $T_{charge-est(k_i)}$ for any node $k_i$ was known in this problem and $t_{01} = 0$, it was possible to calculate the total mission time with Equations (6), (12) and (13), since there were no other variables in Equations (12) and (13).

We defined $T_{charge-est(k_i)}$ in Section 3.2 to denote the required charging time on node $k_i$ if $\rho_{k_i (t)} \equiv 1$. There was a constraint in regard to the total estimated charging time $\sum_{i=1}^{L-2} T_{charge-est(k_i)}$ in this problem. It claimed that the total energy charged during the mission should be the same as the total energy cost during the mission minus the initial energy the UAV took when the mission started:

$$\sum_{i=1}^{L-2} T_{charge-est(k_i)} = \sum_{i=1}^{L-2} (\rho_{k_i (peak)} \cdot \frac{720}{\pi} \sin \frac{\pi(t_{i1}-360)}{720} - \rho_{k_i (peak)} \cdot \frac{720}{\pi} \sin \frac{\pi(t_{i0}-360)}{720})$$
$$= T^{\rho_{max}}_{charge(max)} \times \frac{d_{total}-d_{max}}{d_{max}} \quad (14)$$

where $T^{\rho_{max}}_{charge(max)} \times \frac{d_{total}-d_{max}}{d_{max}}$ shows a constant that shows the total energy needed for charging if a UAV were to fly along path $P_k$. Array $[T_{charge-est(k_1)}, T_{charge-est(k_2)}, \ldots, T_{charge-est(k_{L-2})}]$ represents the estimated charging time sequence in path $P_k$. Additionally, assume all the missions started from t=0, and when the UAVs arrived at the destination, their batteries were always empty (except in the situation where we did not need to charge the UAV during the voyage, where $T^{\rho_{max}}_{charge(max)} \times \frac{d_{total}-d_{max}}{d_{max}} \leq 0$, where

$\sum_{i=1}^{L-2} T_{charge-est(k_i)} = 0$). The total mission time when the UAV left node $k_i$ was defined as $T_{total(i)}$ ($i \in [0, L-1]$). $T_{total-flight(i)}$ describes the total flying time when the UAV left node $k_i$ and $T_{total-charge(i)}$ describes the total charging time when the UAV left node $k_i$.

Besides Equation (14), we had another constraint which described the range of the estimated charging time on each node:

$$\max\left[\left(T^{\rho_{max}}_{charge(max)}\frac{d_{(k_w)\to(k_{w+1})}}{d_{max}} - R_w\right), 0\right] \leq T_{charge-est(k_w)} \leq T^{\rho_{max}}_{charge(max)} \quad (15)$$

where $R_w = \sum_{i=1}^{w} T_{charge-est(k_i)} - \frac{T_{flight-total(w)} - T_{flight(max)}}{T_{flight(max)}} T^{\rho_{max}}_{charge(max)}$ shows the remaining energy when the UAV arrived at nodes $k_w$ in $P_k$. $R_w$ was in minutes, which meant it represented how much time we needed to charge the UAV to the current remaining energy if $\rho_{k_i(t)} \equiv 1$. To make sure we had enough charged energy to reach node $k_{w+1}$, $T_{charge-est(k_w)}$ needed to be larger than $Max\left[\left(T^{\rho_{max}}_{charge(max)}\frac{d_{(k_w)\to(k_{w+1})}}{d_{max}} - R_w\right), 0\right]$. $T_{total(w)}$ represents the total mission time when the UAV arrived at node $k_w$.

Let $[X_1, X_2, \ldots, X_{L-2}]$ denote $\left[T_{charge-est(k_1)}, T_{charge-est(k_2)}, \ldots, T_{charge-est(k_{L-2})}\right]$, $[Y_1, Y_2, \ldots, Y_{L-2}]$ denote $\left[T_{total(1)}, T_{total(2)}, \ldots, T_{total(L-2)}\right]$, $[\rho_1, \rho_2, \ldots, \rho_{L-2}]$ denote $\left[\rho_{k_1(peak)}, \rho_{k_2(peak)}, \ldots, \rho_{k_{L-2}(peak)}\right]$ and $[d_1, d_2, \ldots, d_{L-1}]$ denote $\left[d_{(k_0)\to(k_1)}, d_{(k_1)\to(k_2)}, \ldots, d_{(k_{L-2})\to(k_{L-1})}\right]$. $T_{cm} = T^{\rho_{max}}_{charge(max)}$, $D_m = d_{max}$, $F_m = T_{flight(max)}$ and $F_i = T_{flight(max)}\frac{d_i}{D_m}$.

Then, the minimization problem for the total charging time along path $P_k$ could be described as:

$$\begin{aligned} \min_{[X_1, X_2, \ldots, X_{L-2}] \in [0, T_{cm}]} \quad & Y_{L-1}, \text{ s.t. } Y_{L-1} = Y_{L-2} + F_{L-1}; \; Y_{L-2} \\ & = \left(\frac{720}{\pi}\sin^{-1}\left(\frac{\pi}{720\rho_{L-2}}(x_{L-2}\right.\right. \\ & \left.\left. + \rho_{L-2}\frac{720}{\pi}\sin\frac{\pi(Y_{L-3}+F_{L-2}-360)}{720}\right)\right) + 360\right) \end{aligned} \quad (16)$$

where for any $X_i$ and $X_j$, $\sum_{i=1}^{L-2} X_i = T_{cm}\frac{\sum_{i=1}^{L-1} d_i - D_m}{D_m}$ and $Max\left[\left(T_{cm}\frac{d_j}{D_m} - R_j\right), 0\right] \leq X_j \leq T_{cm}$.

This was a multivariable minimization problem with an iterative expression. Hence, it was quite difficult to find a globally optimal solution for the total mission time minimization problem with dynamic charging efficiency in any $P_k$ path (in [3], such a path planning problem was proved to be an NP-hard problem). Even if we simplified the problem by assigning $[Y_1, Y_2, \ldots, Y_{L-2}]$ to be integers, the problem became an integer factorization problem, which could not be solved in polynomial time. Thus, in Section 5.2, we found a locally optimal solution for Equation (16) instead; then, we proposed a dynamically greedy algorithm to solve the mission time minimization problem.

## 5.2. Locally Optimal CTA

If we only focused on one single iteration of Equation (16), the problem would become much simpler (the total time for one iteration of Equation (16) could be presented as $Y_w = \left(\frac{720}{\pi}\sin^{-1}\left(\frac{\pi}{720\rho_w}(X_w + \rho_w\frac{720}{\pi}\sin\frac{\pi(Y_{w-1}+F_w-360)}{720})\right) + 360\right)$; in such a problem, the iterative expression was removed). In other words, we considered a locally optimal problem to assign the charging time among two adjacent nodes $k_i$ and $k_{i+1}$ in path $P_k$. Assume that when the UAV arrived at node $k_i$, it stored $R_i$ energy, while when it left node $k_i$, it stored $R_i'$ energy. Let $R_{i'max}'$ denote the maximum value that $R_i'$ could be, and let $R_{i'min}'$ denote the minimum value that $R_i'$ could be ($R_{i'min}' = T_{cm}\frac{d_{j+1}}{D_m}$ when $R_i < T_{cm}\frac{d_{j+1}}{D_m}$ while $R_{i'min}' = R_i$ when $R_i \geq T_{cm}\frac{d_{j+1}}{D_m}$). Let $t_{i1(min)}$ denote the time that the UAV achieves $R_{i'min}'$ stored energy on node $k_i$, $t_{i1}$ denote the actual time the UAV leaves node $k_i$ and $t_{i1(max)}$ denote the time

that the UAV achieves $R'_{i_{max}}$ stored energy on node $k_i$. Let $T_{total(i,(i+1))}$ denote the time that the UAV achieves the required additional energy $R'_{i_{store}} = R'_{i_{max}} - R'_{i_{min}}$ on node $k_{i+1}$, where $R'_{i_{max}} \in \left[0, T^{\rho_{max}}_{charge(max)}\right)$ and $R'_{i_{min}} \in [0, R'_{i_{max}}]$.

To calculate the total mission time $T_{total(i,(i+1))}$ for this local CTA problem, we used $t_{i1}$ as the variable instead of $X_i$, because it described the locally optimal problem more precisely. Equation (17) shows the way to calculate $T_{total(i,(i+1))}$ with $t_{i1}$ (obtained by using Equations (11) and (16)):

$$R'_i - R'_{i_{min}} = \rho_i \frac{720}{\pi} \sin \frac{\pi(t_{i1} - 360)}{720} - \rho_i \frac{720}{\pi} \sin \frac{\pi\left(t_{i1(min)} - 360\right)}{720}$$

$$\begin{aligned}
T_{total(i,(i+1))} &= \frac{720}{\pi} \sin^{-1}\left(\frac{\pi}{720\rho_{i+1}}\left(R'_{i_{store}}\right) - \frac{\rho_i}{\rho_{i+1}} \sin \frac{\pi(t_{i1}-360)}{720}\right. \\
&\quad \left. - \frac{\rho_i}{\rho_{i+1}} \sin \frac{\pi\left(t_{i1(min)}-360\right)}{720} + \sin \frac{\pi(t_{i1}+F_{i+1}-360)}{720}\right) + 360
\end{aligned} \tag{17}$$

where $R'_{i_{store}} - \left(R'_i - R'_{i_{min}}\right)$ represents the estimated charging time on node $k_{i+1}$.

Therefore, the locally mission time minimization problem could be described with the following formula using variable $t_{i1}$:

$$\begin{aligned}
\min_{t_{i1} \in [t_{i1(min)}, t_{i1(max)}]} \quad & T_{total(i,(i+1))}, \text{ s.t. } T_{total(i,(i+1))} = \frac{720}{\pi} \sin^{-1}\left(\frac{\pi}{720\rho_{i+1}}\left(R'_{i_{stone}}\right)\right. \\
& \left. - \frac{\rho_i}{\rho_{i+1}} \sin \frac{\pi(t_{i1}-360)}{720} - \frac{\rho_i}{\rho_{i+1}} \sin \frac{\pi\left(t_{i1(min)}-360\right)}{720}\right. \\
& \left. + \sin \frac{\pi(t_{i1}+F_{i+1}-360)}{720}\right) + 360
\end{aligned} \tag{18}$$

where $\max\left[0, \left(T_{cm}\frac{F_i}{F_m} - R_i\right)\right] \leq \rho_i \frac{720}{\pi} \sin \frac{\pi(t_{i1}-360)}{720} - \rho_i \frac{720}{\pi} \sin \frac{\pi\left(t_{i1(min)}-360\right)}{720}$
$\leq (R'_{i_{max}} - R'_{i_{min}})$. $t_{i1(max)} = \left(\frac{720}{\pi} \sin^{-1}\left(\frac{\pi}{720\rho_i}\left((R'_{i_{max}} - R_i) + \rho_i \frac{720}{\pi} \sin \frac{\pi(t_{i0}-360)}{720}\right)\right) + 360\right) =$
$\left(\frac{720}{\pi} \sin^{-1}\left(\frac{\pi}{720\rho_i}\left((R'_{i_{max}} - R'_{i_{min}}) + \rho_i \frac{720}{\pi} \sin \frac{\pi\left(t_{i1(min)}-360\right)}{720}\right)\right) + 360\right)$. (When $R_i < T_{cm}\frac{F_i}{F_{max}}$, $t_{i1(min)} > t_{i0}$, otherwise, $t_{i1(min)} = t_{i0}$.)

After processing a derivation on Equation (17), we had:

$$\frac{T_{total(i,(i+1))}}{dt_{i1}} = \frac{\cos \frac{\pi(t_{i1}+F_{i+1}-360)}{720} - \frac{\rho_i}{\rho_{i+1}} \cos \frac{\pi(t_{i1}-360)}{720}}{\cos\left(\frac{\pi}{720\rho_{i+1}}X_i - \frac{\rho_i}{\rho_{i+1}} \sin \frac{\pi(t_{i1}-360)}{720} - \frac{\rho_i}{\rho_{i+1}} \sin \frac{\pi\left(t_{i1(min)}-360\right)}{720} + \sin \frac{\pi(t_{i1}+F_{i+1}-360)}{720}\right)} \tag{19}$$

In Equation (19), we were able to find that three singular points could possibly be the candidate $t_{i1}$ to minimize the total charging time between nodes $k_i$ and $k_{i+1}$: $t_{i1(min)}$, $t_{i1(max)}$ and the singular point calculated with the following equation with $t_{i1}$ as the variable:

$$\frac{\rho_i}{\rho_{i+1}} \cos \frac{\pi(t_{i1} - 360)}{720} - \cos \frac{\pi(t_{i1} + F_{i+1} - 360)}{720} = 0 \tag{20}$$

The singular point calculated with Equation (20) was:

$$t_{i1(singular)} = \frac{720}{\pi} \tan^{-1} \frac{\cos\left(\frac{\pi(F_{i+1})}{720} - \frac{\rho_i}{\rho_{i+1}}\right)}{\sin \frac{\pi(F_{i+1})}{720}} + 360 \tag{21}$$

**Conclusion 5.1.** *In the locally optimal problem described in Equation (18), when the solution of Equation (21) $t_{i1(singular)}$ was in $[t_{i1(min)}, t_{i1(max)}]$, $T_{total(i,(i+1))}$ had three candidate extreme points, which were $t_{i1(singular)}$, $t_{i1(min)}$ and $t_{i1(max)}$; when the solution of Equation (21) was not in $[t_{i1(min)}, t_{i1(max)}]$, $T_{total(i,(i+1))}$ had two extreme points, which were $t_{i1(min)}$ and $t_{i1(max)}$.*

However, we could not make sure which of the candidate extreme points resulted in $T_{total(i,(i+1))}$ being minimized by the formulas above. Thus, we needed to divide the problem into different situations from 5.1 to 5.3, and further analyze the locally optimal problem.

### 5.2.1. Situation 5.1 $\rho_i > \rho_{i+1}$

$x_1 = t_{i1(min)}$ is the time the UAV needed to charge to have enough energy to reach node $k_{i+1}$; $x_2 = t_{i1}$ is the exact time that the UAV left node $k_i$; $x_3 = x_2 + F_{i+1}$ is the time the UAV arrived at node $k_{i+1}$; $x_4$ is the time the UAV charged for to have enough energy for the stored energy of $R'_{istore}$ to be achieved after a charging process on node $k_{i+1}$ and $x_4 = T_{total(i,(i+1))}$. Thus, using Equation (11), the estimated charging time for the UAV to charge itself from $R'_{imin}$ stored energy to $R'_i$ energy on node $k_i$ could be calculated with the following equation:

$$R'_i - R'_{imin} = \int_{x_1}^{x_2} \rho_{k_i(t)} d_t \tag{22}$$

Additionally, the estimated charging time for the UAV to charge itself with enough stored energy to $R'_{imax}$ on node $k_{i+1}$ could be calculated with Equation (23):

$$R'_{istore} - (R'_i - R'_{imin}) = (R'_{imax} - R'_{imin}) - (R'_i - R'_{imin}) = (R'_{imax} - R'_i) = \int_{x_3}^{x_4} \rho_{k_{i+1}(t)} d_t \tag{23}$$

where $R'_{istore} - (R'_i - R'_{imin})$ represents how much energy was still needed to charge on node $k_{i+1}$ to achieve the estimated stored energy $R'_{istore}$.

Combining Equations (22) and (23), we had:

$$R'_{istore} = (R'_{imax} - R'_{imin}) = \int_{x_1}^{x_2} \rho_{k_i(t)} d_t + \int_{x_3}^{x_4} \rho_{k_{i+1}(t)} d_t \tag{24}$$

Assume in plan A that the UAV leaves node $k_i$ at time $x_2$, and then plan B where the UAV leaves node $k_i$ at time $x_2{}^*$. In plan B, $x_3{}^* = x_2{}^* + F_{i+1}$ is the time the UAV arrives at node $k_{i+1}$ and $x_4{}^*$ is the time the UAV takes to charge to have enough stored energy of $R'_{istore}$. $x_2{}^* - x_2 = x_3{}^* - x_3$, since we had $x_3 = x_2 + F_{i+1}$ and $x_3{}^* = x_2{}^* + F_{i+1}$.

**Remark 5.1.** *In Figure 8, if $\rho_i > \rho_{i+1}$, $x_2 < 360$ and $x_3 < 360$, and $t_{i1(singular)} \in [t_{i1(min)}, t_{i1(max)}]$. We assumed in plan A that $x_2 = t_{i2(singular)}$. Then, in plan B, when $360 \geq x_2{}^* \geq t_{i1(singular)}$, we could conclude that $x_4{}^* < x_4$.*

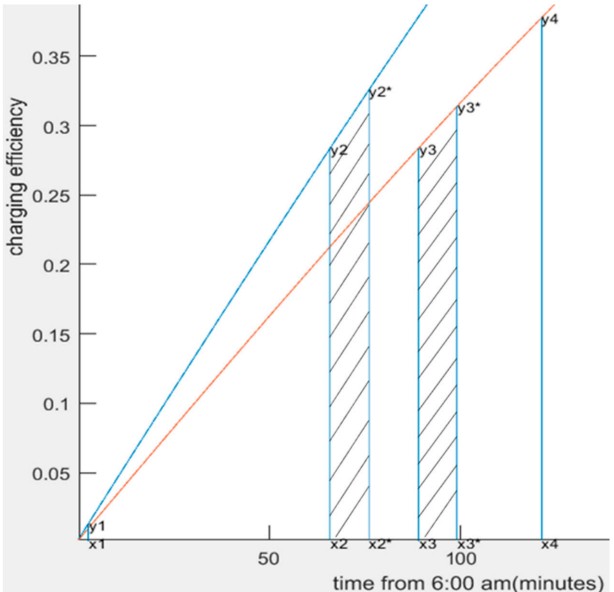

**Figure 8.** Example when $\rho_i > \rho_{i+1}$, $x_2 < 360$ and $x_3 < 360$ ($\rho_i = 1$, $\rho_{i+1} = 0.75$, $F_{i+1} = 23$).

**Proof of Remark 5.1.** $\frac{(\rho_i \cos \frac{\pi(x-360)}{720})}{d_x} = \frac{\pi\rho_i}{720} \sin \frac{\pi(x-360)}{720}$, $\frac{(\rho_{i+1} \cos \frac{\pi(x+F_{i+1}-360)}{720})}{d_x} = \frac{\pi\rho_{i+1}}{720}$ $\sin \frac{\pi(x+F_{i+1}-360)}{720}$, $x + F_{i+1} < 360$ and $\rho_i > \rho_{i+1}$. Then, $\frac{(\rho_i \cos \frac{\pi(x-360)}{720})}{d_x} > \frac{(\rho_{i+1} \cos \frac{\pi(x+F_{i+1}-360)}{720})}{d_x}$ could be concluded easily. In Figure 8, assume that $x_2{}^*$ is quite close to $x_2$; then, $x_3{}^*$ is quite close to $x_3{}^*$ as well. To make $\frac{x_4}{d_{x_2}} = 0$, we had $x_4 = x_4{}^*$. Then, to make $x_4 = x_4{}^*$, the bar area formed by four points $(x_2, 0)$, $(x_2{}^*, 0)$, $(x_2, \rho_{k_i(x_2)})$, $(x_2{}^*, \rho_{k_i(x_2{}^*)})$, which can be represented as $\int_{x_2}^{x_2{}^*} \rho_{k_i(t)} d_t$ and the bar area formed by four points $(x_3, 0)$, $(x_3{}^*, 0)$, $(x_3, \rho_{k_{i+1}(x_3)})$, $(x_3{}^*, \rho_{k_{i+1}(x_3{}^*)})$, which can be represented as $\int_{x_3}^{x_3{}^*} \rho_{k_{i+1}(t)} d_t$, should have the same area. Thus, if $x_2 = t_{i1(\sin gular)}$, we had $\int_{x_2}^{x_2{}^*} \rho_{k_i(t)} d_t = \int_{x_3}^{x_3{}^*} \rho_{k_{i+1}(t)} d_t$ when $x_2{}^*$ was quite close to $x_2$. It is obvious that only when $\rho_{k_i(x_2)} = \rho_{k_{i+1}(x_3)}$ did $x_2 = t_{i1(\sin gular)}$ hold. Hence, if $x_2 = t_{i1(\sin gular)}$, $\rho_{k_i(x_2)} = \rho_{k_{i+1}(x_3)}$. $\square$

The following equation shows the relationship between plan A and plan B:

$$\int_{x_1}^{x_2} \rho_{k_i(t)} d_t + \int_{x_3}^{x_4} \rho_{k_{i+1}(t)} d_t$$
$$= \int_{x_1}^{x_2} \rho_{k_i(t)} d_t + \int_{x_2}^{x_2{}^*} \rho_{k_i(t)} d_t + (\int_{x_3}^{x_4} \rho_{k_{i+1}(t)} d_t \qquad (25)$$
$$- \int_{x_3}^{x_3{}^*} \rho_{k_{i+1}(t)} d_t) + \int_{x_4}^{x_4{}^*} \rho_{k_{i+1}(t)} d_t$$

With Equation (25), we could conclude that if $\int_{x_2}^{x_2{}^*} \rho_{k_i(t)} d_t > \int_{x_3}^{x_3{}^*} \rho_{k_{i+1}(t)} d_t$, then $x_4{}^* < x_4$.

Then, if we compared $\int_{x_2}^{x_2{}^*} \rho_{k_i(t)} d_t$ and $\int_{x_3}^{x_3{}^*} \rho_{k_{i+1}(t)} d_t$, since $\rho_{k_i(x_2)} = \rho_{k_{i+1}(x_3)}$ and $\frac{(\rho_i \cos \frac{\pi(x_2{}^*-360)}{720})}{d_{x_2{}^*}} > \frac{(\rho_{i+1} \cos \frac{\pi(x_2{}^*+F_i-360)}{720})}{d_{x_2{}^*}}$, we could conclude that $\int_{x_2}^{x_2{}^*} \rho_{k_i(t)} d_t > \int_{x_3}^{x_3{}^*} \rho_{k_{i+1}(t)} d_t$ when $(x_2{}^* - x_2)$ was not a quite small value.

**Remark 5.2.** *If $\rho_i > \rho_{i+1}$, $x_2 < 360$ and $x_3 < 360$, and $t_{i1(singular)} \in [t_{i1(min)}, t_{i1(max)}]$. Assume that $x_2 = t_{i1(singular)}$ and $x_2{}^* < x_2$; similarly, $\int_{x_2{}^*}^{x_2} \rho_{k_i(t)} d_t > \int_{x_3{}^*}^{x_3} \rho_{k_{i+1}(t)} d_t$. Thus, $x_4{}^* < x_4$. (The proof is like Remark 5.1.)*

**Remark 5.3.** *If $\rho_i > \rho_{i+1}$, $x_2 < 360$ and $x_3 < 360$, and $t_{i1(singular)} \in [t_{i1(min)}, t_{i1(max)}]$, $T_{total(i,(i+1))}$ is a convex function in range $[t_{i1(min)}, t_{i1(max)}]$ due to Remarks 5.1 and 5.2. Thus, $T_{total(i,(i+1))}$ should be compared when $t_{i1} = t_{i1(min)}$ and $t_{i1} = t_{i1(max)}$ and find the result with the minimum total charging time.*

**Proof of Remark 5.3.** If $\rho_i > \rho_{i+1}$, $x_2 < 360$ and $x_3 < 360$, when $x_2 = t_{i1(\sin gular)}$ and $t_{i1(\sin gular)} \in [t_{i1(min)}, t_{i1(max)}]$, we always had $x_4{}^* < x_4$. Thus, the result of $x_4{}^*$ obtained a maximum value in plan A. Using Remarks 5.1 and 5.2, $\frac{x_4{}^*}{d_{x_2{}^*}} > 0$, when $x_2{}^* < x_2$; $\frac{x_4{}^*}{d_{x_2{}^*}} < 0$, when $x_2{}^* > x_2$. ($x_4{}^*$ had only one singular point, where $\frac{x_4{}^*}{d_{x_2{}^*}} = 0$, and it decreased either when $x_2{}^* < x_2$ or $x_2{}^* > x_2$). $\square$

**Remark 5.4.** *If $\rho_i > \rho_{i+1}$, $x_2 < 360$ and $x_3 < 360$, and $t_{i1(singular)} \notin [t_{i1(min)}, t_{i1(max)}]$, $t_{i1(singular)}$ must be less than $t_{i1(min)}$ or larger than $t_{i1(max)}$. If $t_{i1(singular)} > t_{i1(max)}$, assume $x_2 = t_{i1(max)}$, since $\frac{(\rho_i \cos \frac{\pi(x-360)}{720})}{d_x} > \frac{(\rho_{i+1} \cos \frac{\pi(x+F_{i+1}-360)}{720})}{d_x}$, $\rho_{k_i(x_2)} < \rho_{k_{i+1}(x_3)}$. Additionally, since $t_{i1(singular)} \notin [t_{i1(min)}, t_{i1(max)}]$, $x_4{}^*$ is a monotonic function by $x_2{}^*$. Then, also by $\frac{(\rho_i \cos \frac{\pi(x-360)}{720})}{d_x} > \frac{(\rho_{i+1} \cos \frac{\pi(x+F_{i+1}-360)}{720})}{d_x}$, $t_{i1} = t_{i1(min)}$ obtains a minimum $x_4{}^*$. Moreover, if $t_{i1(singular)} < t_{i1(min)}$, assume $x_2 = t_{i1(min)}$, since $\frac{(\rho_i \cos \frac{\pi(x-360)}{720})}{d_x} > \frac{(\rho_{i+1} \cos \frac{\pi(x+F_i-360)}{720})}{d_x}$, $\rho_{k_i(x_2)} > \rho_{k_{i+1}(x_3)}$. Additionally, since $t_{i1(singular)} \notin [t_{i1(min)}, t_{i1(max)}]$, $x_4{}^*$ is a monotonic func-*

tion by $x_2{}^*$. Then, also by $\frac{(\rho_i \cos \frac{\pi(x-360)}{720})}{d_x} > \frac{(\rho_{i+1} \cos \frac{\pi(x+F_{i+1}-360)}{720})}{d_x}$, $x_2{}^* = t_{i1} = t_{i1(max)}$ obtains a minimum $x_4{}^*$.

**Proof of Remark 5.4.** Additionally, in Figure 8, if we assumed $t_{i1(\sin gular)} > t_{i1(max)}$ and $x_2 = t_{i1(max)}$, then we had $x_2{}^* < x_2$, $\rho_{k_i(x_2)} < \rho_{k_{i+1}(x_3)}$ and $\frac{(\rho_i \cos \frac{\pi(x-360)}{720})}{d_x} > \frac{(\rho_{i+1} \cos \frac{\pi(x+F_{i+1}-360)}{720})}{d_x}$. Thus, $\int_{x_2{}^*}^{x_2} \rho_{k_i(t)} d_t < \int_{x_3{}^*}^{x_3} \rho_{k_{i+1}(t)} d_t$; then, we had $\int_{x_1}^{x_2} \rho_{k_i(t)} d_t + \int_{x_3}^{x_4} \rho_{k_{i+1}(t)} d_t < \int_{x_1}^{x_2{}^*} \rho_{k_i(t)} d_t + \int_{x_3{}^*}^{x_4} \rho_{k_{i+1}(t)} d_t$ (if we still charged the UAV on node $k_{i+1}$ till $x_4$ and we changed the time the UAV left node $k_i$ from $x_2$ to $x_2{}^*$, $x_2{}^* < x_2$, the stored energy could extend $R'_{i\text{store}}$. Thus, it would take less time than $x_4$ to charge the UAV to $R'_{i\text{store}}$). Hence, $x_4{}^* < x_4$. Since $x_4{}^*$ is a monotonic function by $x_2{}^*$, $x_4{}^*$ obtains a minimized value on $x_2{}^* = t_{i1(min)}$. Similarly, if $t_{i1(\sin gular)} < t_{i1(min)}$, when $x_2{}^* = t_{i1} = t_{i1(max)}$, $x_4{}^*$ would obtain a minimized value. $\square$

**Remark 5.5.** *Similarly to Figure 8, if $\rho_i > \rho_{i+1}$, $x_3 \geq 360$, $x_2 < 360$, we would have a similar conclusion to Remarks 5.3 and 5.4: when $\rho_i > \rho_{i+1}$, if $t_{i1(singular)} \in [t_{i1(min)}, t_{i1(max)}]$, the result of should be compared when $t_{i1} = t_{i1(min)}$ and $t_{i1} = t_{i1(max)}$, and the result with the minimum total charging time should be found, while if $t_{i1(singular)} \notin [t_{i1(min)}, t_{i1(max)}]$, when $t_{i1(singular)} > t_{i1(max)}$ $t_{i1} = t_{i1(min)}$ has the minimum total charging time, and when $t_{i1(singular)} < t_{i1(min)}$, $t_{i1} = t_{i1(max)}$ has the minimum total charging time. (It can be easily proved with the same steps as those in Remarks 5.1.–5.4).*

**Remark 5.6.** *In situation 5.1, if $x_2 > 360$, $t_{i1(singular)}$ could not be larger than 360 when $\rho_i > \rho_{i+1}$. Thus, it is obvious $t_{i1} = t_{i1(max)}$ was the solution of Equation (17) under this situation.*
*Proof of Remark 5.6 $\rho_{k_i(x_2)} > \rho_{k_{i+1}(x_2)}$, $\rho_{k_{i+1}(x_2)} > \rho_{k_{i+1}(x_3)}$, thus $\rho_{k_i(x_2)} > \rho_{k_{i+1}(x_3)}$, similarly, $\rho_{k_i(x_2{}^*)} > \rho_{k_{i+1}(x_3{}^*)}$. Thus, $\int_{x_2}^{x_2{}^*} \rho_{k_i(t)} d_t > \int_{x_3}^{x_3{}^*} \rho_{k_{i+1}(t)} d_t$. Hence, $x_4{}^* < x_4$, and the more we charged the UAV on node $k_i$, the less total charging time we obtained.*

**Operation 5.1.** *When $\rho_i > \rho_{i+1}$, if $t_{i1(singular)} \in [t_{i1(min)}, t_{i1(max)}]$, we compared $T_{total(i,(i+1))}\big|t_{i1} = t_{i1(min)}$ and $T_{total(i,(i+1))}\big|t_{i1} = t_{i1(max)}$ and chose the CTA plan with a smaller $T_{total(i,(i+1))}$. If $t_{i1(singular)} > t_{i1(max)}$, we chose $t_{i1} = t_{i1(min)}$ to minimize $T_{total(i,(i+1))}$, while if $t_{i1(singular)} < t_{i1(min)}$, we chose $t_{i1} = t_{i1(max)}$ to minimize $T_{total(i,(i+1))}$.*

5.2.2. Situation 5.2 $\rho_i = \rho_{i+1}$

**Remark 5.7.** *When $\rho_i = \rho_{i+1}$, if $t_{i1(singular)} \in [t_{i1(min)}, t_{i1(max)}]$, $360 - t_{i1(singular)} = \frac{F_{i+1}}{2}$. Assume $x_2 = t_{i1(singular)}$, when $x_2{}^* < x_2$, we had $x_4{}^* < x_4$. When $x_2{}^* > x_2$, we also had $x_4{}^* < x_4$. Thus, $T_{total(i,(i+1))}$ is a convex function in range $[t_{i1(min)}, t_{i1(max)}]$, the same as situation 5.1. Hence, if we compared $T_{total(i,(i+1))}\big|t_{i1} = t_{i1(min)}$ and $T_{total(i,(i+1))}\big|t_{i1} = t_{i1(max)}$, we found the minimized $T_{total(i,(i+1))}$. (The proof is like Remarks 5.1.–5.3).*

**Remark 5.8.** *When $\rho_i = \rho_{i+1}$, if $t_{i1(singular)} \notin [t_{i1(min)}, t_{i1(max)}]$, and if $t_{i1(max)} < t_{i1(singular)}$, $T_{total(i,(i+1))}\big|t_{i1} = t_{i1(min)}$ resulted in a minimized value of $T_{total(i,(i+1))}$; if $t_{i1(min)} > t_{i1(singular)}$, $T_{total(i,(i+1))}\big|t_{i1} = t_{i1(max)}$ resulted in a minimized value of $T_{total(i,(i+1))}$. (The proof is like Remark 5.4.)*

**Operation 5.2.** *When $\rho_i = \rho_{i+1}$, if $t_{i1(singular)} \in [t_{i1(min)}, t_{i1(max)}]$, we compared $T_{total(i,(i+1))}\big|t_{i1} = t_{i1(min)}$ and $T_{total(i,(i+1))}\big|t_{i1} = t_{i1(max)}$ and chose the CTA plan with a smaller $T_{total(i,(i+1))}$. If $t_{i1(singular)} > t_{i1(max)}$, we chose $t_{i1} = t_{i1(min)}$ to minimize $T_{total(i,(i+1))}$, while if $t_{i1(singular)} < t_{i1(min)}$, we chose $t_{i1} = t_{i1(max)}$ to minimize $T_{total(i,(i+1))}$.*

### 5.2.3. Situation 5.3 $\rho_i < \rho_{i+1}$

**Remark 5.9.** *Like Remark 5.6, if $\rho_i < \rho_{i+1}$ and $x_3 < 360$, then $t_{i1(singular)} \notin [t_{i1(min)}, t_{i1(max)}]$. Since $\int_{x_2}^{x_2^*} \rho_{k_i(t)} d_t < \int_{x_3}^{x_3^*} \rho_{k_{i+1}(t)} d_t$, $t_{i1} = t_{i1(min)}$ resulted in a minimized total charging time $T_{total(i,(i+1))}$.*

**Remark 5.10.** *If $\rho_i < \rho_{i+1}$ and $x_2 > 360$, assume $x_2 = t_{i1(singular)}$, $t_{i1(singular)} \in [t_{i1(min)}, t_{i1(max)}]$, $T_{total(i,(i+1))}$ is a convex function in range $[t_{i1(min)}, t_{i1(max)}]$. Thus, if we compared $T_{total(i,(i+1))}\big|t_{i1} = t_{i1(min)}$ and $T_{total(i,(i+1))}\big|t_{i1} = t_{i1(max)}$, we found the minimized $T_{total(i,(i+1))}$. If $t_{i1(max)} < t_{i1(singular)}$, $T_{total(i,(i+1))}\big|t_{i1} = t_{i1(min)}$ resulted in a minimized value of $T_{total(i,(i+1))}$; if $t_{i1(min)} > t_{i1(singular)}$, $T_{total(i,(i+1))}\big|t_{i1} = t_{i1(max)}$ resulted in a minimized value of $T_{total(i,(i+1))}$.*

**Remark 5.11.** *In remark 5.10, if $x_2 < 360$ and $x_3 > 360$, we obtained the same conclusion as Remark 5.10. In conclusion, when $x_3 > 360$, if $t_{i1(singular)} \in [t_{i1(min)}, t_{i1(max)}]$, and if we compared $T_{total(i,(i+1))}\big|t_{i1} = t_{i1(min)}$ and $T_{total(i,(i+1))}\big|t_{i1} = t_{i1(max)}$, we could find the minimized $T_{total(i,(i+1))}$. If $t_{i1(max)} < t_{i1(singular)}$, $T_{total(i,(i+1))}\big|t_{i1} = t_{i1(min)}$ resulted in a minimized value of $T_{total(i,(i+1))}$; if $t_{i1(min)} > t_{i1(singular)}$, $T_{total(i,(i+1))}\big|t_{i1} = t_{i1(max)}$ resulted in a minimized value of $T_{total(i,(i+1))}$.*

**Operation 5.3.** *When $\rho_i < \rho_{i+1}$, if $t_{i1(singular)} \in [t_{i1(min)}, t_{i1(max)}]$, we compared $T_{total(i,(i+1))}\big|t_{i1} = t_{i1(min)}$ and $T_{total(i,(i+1))}\big|t_{i1} = t_{i1(max)}$ and chose the CTA plan with a smaller $T_{total(i,(i+1))}$. If $t_{i1(singular)} > t_{i1(max)}$, we chose $t_{i1} = t_{i1(min)}$ to minimize $T_{total(i,(i+1))}$, while if $t_{i1(singular)} < t_{i1(min)}$, we chose $t_{i1} = t_{i1(max)}$ to minimize $T_{total(i,(i+1))}$.*

**Operation 5.4.** *In different situations, if $t_{i1(singular)} \in [t_{i1(min)}, t_{i1(max)}]$, we compared $T_{total(i,(i+1))}\big|t_{i1} = t_{i1(min)}$ and $T_{total(i,(i+1))}\big|t_{i1} = t_{i1(max)}$ and chose the CTA plan with a smaller $T_{total(i,(i+1))}$. If $t_{i1(singular)} > t_{i1(max)}$, we chose $t_{i1} = t_{i1(min)}$ to minimize $T_{total(i,(i+1))}$, while if $t_{i1(singular)} < t_{i1(min)}$, we chose $t_{i1} = t_{i1(max)}$ to minimize $T_{total(i,(i+1))}$.*

### 5.3. Dynamically Greedy CTA Algorithm

Section 5.2 shows all the situations we were met with in the locally optimal CTA problem described by Equation (18). $t_{i1(min)}$ could be calculated in any known path P. However, $t_{i1(max)}$ was not determined in the locally optimal problem. If we applied the CTA algorithm to this problem and used the peak charging efficiency in each node as the statically charging efficiency, we could predict a proper $t_{i1(max)}$.

$\left[ T_{charge-est(k_1)}, T_{charge-est(k_2)}, \cdots, T_{charge-est(k_{L-2})} \right]$ (shown in Figure 9) could describe different charging time assignment plans obtained in this problem, as follows in the image shown.

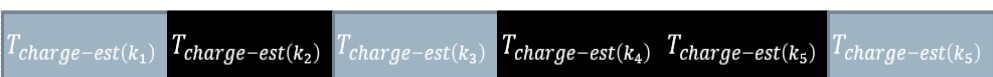

**Figure 9.** Charging time assignment example using Algorithm 1.

The grey bar means we only charged enough power on the current node to reach the next node, while the black bar means we charged the UAV to a full battery state. If we only modified the two adjacent CTAs $T_{charge-est(k_i)}$ and $T_{charge-est(k_{i+1})}$, and kept $T_{charge-est(k_i)} + T_{charge-est(k_{i+1})}$ unchanged, the CTA problem among node $k_i$ and $k_{i+1}$ became a locally optimal problem, as discussed in Section 5.2. Combining Algorithm 1 and Operation 5.4, we proposed a dynamically greedy CTA(DG-CTA) algorithm as an advanced greedy algorithm. The DG-CTA algorithm focuses on CTA problems with dynamically

charging efficiency. Please note that if the two adjacent nodes were both assigned charging times to charge the UAV to a full battery state, or both to charge the UAV only enough power to reach the next node, we could not apply any changes, since there was only one CTA plan for the current $T_{charge-est(k_i)} + T_{charge-est(k_{i+1})}$.

Algorithm 3 shows the DG-CTA algorithm which improved Algorithm 1 regarding its dynamically charging efficiency. Assume $t_{i1}\prime$ is the new operated time that the UAV left node $k_i$ at.

---

**Algorithm 3**: DG-CTA algorithm.

---

Input: Path $P_k$, graph G, peak charging efficiency $\rho_i$ for all the nodes in $V_{local}$
Output: Total charging time along $P_k$ and the arrival time/leaving time for each node
Perform Algorithm 1 and find all stayed nodes in path $P_k$ as a new path $P_k\prime$ (L—length $(P_k')$),
while all the possible unpruned paths are generated

1.　For i = 1:L-2,
2.　　　　If $\rho_i' < \rho_{i+1}'$, //Situation 5.2.3 is solved with Operation 5.4
3.　　　　　　If $t_{i1(\sin gular)}\prime \in \left[ t_{i1(min)}', \ t_{i1(max)}' \right]$ and

　　　　　　　　$T_{total(i,(i+1))}\prime \Big| t_{i1}' = t_{i1(min)}' > T_{total(i,(i+1))}\prime \Big| t_{i1}\prime = t_{i1(min)}\prime >$
4.　　　　　　　　$t_{i1}\prime = t_{i1(max)}\prime.$
5.　　　　　Else, if $t_{i1(\sin gular)}\prime < t_{i1(min)}' \ t_{i1}\prime = t_{i1(max)}\prime.$
6.　　　　　　　　$t_{i1}\prime = t_{i1(max)}\prime$
7.　　　　　Else, if $t_{i1(\sin gular)}' > t_{i1(max)}\prime$
8.　　　　　　　　$t_{i1}\prime = t_{i1(min)}\prime$
9.　　　　　End
10.　　　　Else, if $\rho_i' > \rho_{i+1}\prime$, //Situation 5.2.3 is solved with Operation 5.4
11.　　　　　　If $t_{i1(\sin gular)}\prime \in [t_{i1(min)}\prime, \ t_{i1(max)}'$ and

　　　　　　　　$T_{total(i,(i+1))}\prime \Big| t_{i1}' = t_{i1(min)}' < T_{total(i,(i+1))}\prime \Big| t_{i1}\prime = t_{i1(max)}\prime$
12.　　　　　　　　$t_{i1}' = t_{i1(min)}'$
13.　　　　　Else, if $t_{i1(\sin gular)}\prime < t_{i1(min)}'$
14.　　　　　　　　$t_{i1}' = t_{i1(max)}\prime$
15.　　　　　Else, if $t_{i1(\sin gular)}' > t_{i1(max)}\prime$
16.　　　　　　　　$t_{i1}\prime = t_{i1(min)}\prime$
17.　　　　　End
18.　　　　Else //Situation 5.2.2 is solved with Operation 5.4
19.　　　　　　If $t_{i1(\sin gular)}\prime \in [t_{i1(min)}\prime, \ t_{i1(max)}\prime$ and

　　　　　　　　$T_{total(i,(i+1))}\prime \Big| t_{i1}\prime = t_{i1(min)}\prime > T_{total(i,(i+1))}\prime \Big| t_{i1}\prime = t_{i1(max)}\prime$
20.　　　　　　　　$t_{i1}\prime = t_{i1(max)}\prime$
21.　　　　　Else, if $t_{i1(\sin gular)}\prime < t_{i1(min)}'$
22.　　　　　　　　$t_{i1}' = t_{i1(max)}\prime$
23.　　　　　Else, if $t_{i1(\sin gular)}' > t_{i1(max)}\prime$
24.　　　　　　　　$t_{i1}\prime = t_{i1(min)}\prime$
25.　　　　　End
26.　　　End
27.　Return $\left[ t_{10}', \ t_{20}', \ \ldots, \ t_{(L-2)0}' \right], \ \left[ t_{11}', \ t_{21}', \ \ldots, \ t_{(L-2)1}' \right]$

---

Algorithm 3 was applied to find the optimal charging time assignment plan in any dynamically charging efficiency environment and could be tested by comparing the total charging time it found with what the other algorithms found. Algorithm 3 was in O(n). However, since the result of Algorithm 1 was required in Algorithm 3, the actual running time of Algorithm 3 was in O$(n^2)$.

### 5.4. Heuristic Algorithm Based on DG-CTA

Although Algorithm 3 did not take too much time compared with the discrete enumerate algorithm (which enumerated the minute assignment plan for the CTA and could not be solved in polynomial time), the traversal of paths using the DFS algorithm combined with

the CTA algorithms was still time-consuming, because Lemma 4.1 could not be applied in this scheme. To further reduce the total running time of the algorithm, we proposed a heuristic algorithm for the SPU path planning problem with dynamically charging efficiency. The purpose of this heuristic algorithm was to quickly find candidate optimal paths with smaller distances and larger averaged charging efficiencies in the early path generation process. The first $NOP_{max}$ paths with the smallest cost among all possible paths were chosen to be put in the candidate path set. The best path we found with the heuristic algorithm was the one with a minimum total mission time among all the paths in the candidate path set.

The cost function of the heuristic algorithm was:

$$h(a) = 2(\gamma_1 T_{total-flight(est)} + \gamma_2 T_{charge(max)}^{\rho_{max}} \frac{T_{total-flight(est)} - T_{flight(max)}}{\rho_{a(k)} T_{flight(max)}}) \qquad (26)$$

where $\rho_{a(k)}$ is the averaged charging efficiency on path $P_k$. $\gamma_1$ and $\gamma_2$ are the weight factors for the heuristic cost function and $\gamma_1 + \gamma_2 = 1$.

Algorithm 4 shows the detail of this heuristic algorithm:

---

**Algorithm 4**: DG-CTA-based heuristic path planning algorithm.

---

Input: Path $P_k$, graph G, peak charging efficiency $\rho_{k_i(peak)}$ for all nodes in $V_{local}$
$T_{total(MEFT)} = T_{total(Dijkstra)}$, $NOP_{max}$ which represents the number of candidate paths
Output: Minimized total mission time $T_{total}$ and optimal path $P_{optimal}$
Variables: a, $P_k$, i
1.   Run DFS path generation algorithm on path $P_k$, calculate the cost for each path using
     Equation (26)
2.   Sort all possible paths by the cost and find the first $NOP_{max}$ paths as the candidate path set
3.   //Build the candidate path set
4.   For i = 1 : $NOP_{max}$
5.   Calculate $T_{total(a)}$ for i$^{th}$ path in the candidate path set using Algorithm 3
6.             If $T_{total(a)} > T_{total(MEFT)}$,
7.                      $P_k$ is pruned.
8.           Else,
9.                      $T_{total(MEFT)} = T_{total(a)}$ and $P_{optimal} = P_k$
10.             End
11.          End
12.  End//calculate mission time for paths
13.  Return $T_{total(MEFT)}$ and $P_{optimal}$

---

Algorithm 4 was applied to find the optimal path for the SPU delivery problem in the dynamically charging efficiency environment, and could be tested by comparing the total mission time it found with what other algorithms found. Since Equation (26) required a running time in $O(n)$, the cost and path generation process required a running time of $O(n^3)$. Thus, let the sorting process take $O(n^2)$, so the total time cost for the candidate path set building process is in $O(n^3)$. To find the path with a minimized mission time, it required a running time of $O(NOP_{max} \times n^2)$. Since $NOP_{max}$ was a constant in this problem, the total running time of Algorithm 4 was in $O(n^3)$. As shown in Section 8.2, the performance of our proposed algorithm was much better than any other existing algorithms (Algorithms 3 and 4 can only be used in solar-powered UAV systems, since the optimization analysis in this section was performed on a solar-powered UAV path planning model).

## 6. UAV Delivery Mission Arrangement for Urban City

Finally, with the single SPU path planning problem solved, we optimized the problem described in Equation (7). In our study, the city map was divided into different local city maps based on the service area of the stores. Thus, we did not find nearby UAVs as [2] was able to.

However, in practical UAV delivery systems, the landing places on a city map may not always be available for the following reasons:

1. The landing spaces may be fully occupied.
2. The landing space may not be open at certain times.
3. Bad weather may be forecasted.

To manage more than one UAV delivery mission, we needed to consider the factors above. Aiming at the first reason, we needed to give the UAV missions different ranks to determine which UAV was chosen to take the limited space in each crowded landing place. Alongside the problems above, we also needed to update the city map and rerun the path planning algorithm in special cases.

*6.1. UAV Mission Arrangement Protocol to Solve Space Limitation*

We proposed a rank-based protocol in our SPU delivery system to solve the space limitation problem in landing places. Before introducing the algorithms and protocols, we needed to initialize the landing places' occupation status $SO(city) = \{SO_1, SO_2, \ldots, SO_n\}$ and the UAV mission rank status $SM(city) = \{SM_1, SM_2, \ldots, SM_m\}$. n is the number of nodes in the local city map and m is the number of missions for the current daytime. $SO_i \leq L_{SO}$, where $L_{SO}$ is the maximum number of UAVs that could stay in a landing place (the existing technique from [21] allowed UAVs to recognize different landing points where different specific landing points were drawn in different shapes or colors). The SO(city) is a real-time function sequence, such as $SO(city) = \{SO_1(t), SO_2(t), \ldots, SO_n(t)\}$. $SO_i(t) = 0$ means that there were no UAV missions that decided to land on it. $SM_j$ represents the current rank of mission j. Assuming we had an initialized estimated arriving time for all missions of $T_{MP(initial)} = \{T_{MP1}, T_{MP2}, \ldots, T_{MPm}\}$, and at the end of each stop of any UAV, we updated a new estimated arriving time to be $T_{MP(realtime)} = \left\{T_{MP(realtime)1}, T_{MP(realtime)2}, \ldots, T_{MP(realtime)m}\right\}$, $SM_{i(initial)} = T_{MP1}$. The rank of the mission was calculated using Equation (27):

$$SM_{i(update)} = \tau_1 T_{MP1} + \tau_2 \left(T_{MP1} - T_{MP(realtime)1}\right) \tag{27}$$

where $\tau_1$ and $\tau_2$ are parameters which balanced the contributions for ranks with the estimated total mission time and the customers' additional waiting time, $\tau_1 + \tau_2 = 1$. Missions with a higher rank had higher weight to charge on a crowded landing place.

Then, we proposed the following delivery mission arrangement protocol to deal with the space limitation problem in landing places:

Delivery mission arrangement protocol:

1. First, we applied Algorithm 4 to each mission and built a database of $SO(city) = \{SO_1(t), SO_2(t), \ldots, SO_n(t)\}$, $T_{MP(initial)} = \{T_{MP1}, T_{MP2}, \ldots, T_{MPm}\}$ and the path information $P_{MP(initial)} = \{P_{MP1}, P_{MP2}, \ldots, P_{MPm}\}$.
2. For any occupation function of $SO_i(t)$, if in any $SO_i(t)$ we had $SO_i(t) > C$, C was the capacity of the landing place. Then, we needed to arrange the missions that would arrive at the node i. We then had to find missions C that had the highest weight and kept their fight mission plan unchanged. Assume $SO_i(t) = K$.
3. For missions with lower weights, other than the first C missions, we ordered them by their mission weights. Choosing missions from a higher weight to lower weight, we reran Algorithm 4 and pruned the current path $P_{MPj}$, then found other paths until a path did not pass node I, or when during the landing time of a newly generated path, node i was not fully occupied.
4. Once the path plan for mission j was updated, we updated $SM_j$, $SO_i(t)$, $T_{MP(realtime)j}$. If no possible path could be found for mission j, $T_{MP1} = T_{MP1} + 720$ (minutes), and it was delayed to the next day.
5. We returned back to step three and changed the mission path plan for the next mission.

6. After we ensured $SO_i(t) \leq C$ at all times of day, we went back to step two and changed the mission path plans for the mission in the next node.

With the rank of the mission introduced, the spaces for landing places could be well scheduled without any conflicts.

### 6.2. Local City Maps Updating Protocol

The algorithms we proposed before was aimed at local city maps, which are different from the integrated city map. While initializing the integrated city map and the local city maps, we considered each partitioned region (local city map) to be the same as the service region for stores by their delivery companies. However, in the following cases, we needed to update the region partition for stores in a city:

1. Some nodes in a local city map could not be reached by the store in charge of that region.
2. It always took less time to reach a node from other stores than the store serving the node's region.
3. Some nodes were not available, and the owners reported the situations.

Hence, we proposed the following strategy as a protocol to update the local city map:

1. When a mission could not be completed because a landing place could not be reached, we calculated whether stores from the adjacent service regions could reach that landing place, and then updated the local city map of the current store and the store that could reach that node with a minimized time.
2. A simulation program wan run to calculate the mission time for each node to fulfil the delivery from a store in the current local city map and other adjacent local city maps. Suppose that node a is currently in a region with service covered by store A, and store B has a shortest mission time to reach the node compared with all other adjacent stores; in that case, this node should be removed from the local city map for store A and added to the local city map for store B.
3. When any unavailability of landing places was reported, the local city map temporarily removed it and added it back when it was available again.

### 6.3. Real-Time Rerouting Protocol

In our proposed SPU delivery system, when a bad weather prediction report was received, the missions related to the corresponding region needed to be rerouted to avoid accidents or to reduce the total mission time.

A bad weather report included the following three conditions:

1. The cloudy level exceeds the threshold of the UAV charging process taking a much longer time.
2. It is raining in this region and the UAVs are not able to fly.
3. The wind speed level exceeds the threshold that UAVs can fly safely with payloads.

When these bad weather reports were received, the related UAVs were rerouted. The landing places these UAVs currently stayed on or the next landing places they would have arrived at could be the start node of a new routing problem. In emergency cases, we applied Algorithm 4 with a smaller $NOP_{max}$ to quickly reroute the UAVs.

With the proposed protocols in this section, the delivery missions in the city map were well organized and the problems we were met with in the system could be solved.

## 7. SPU Delivery Feasibility Study

Before we started the experiments to test the path planning algorithms proposed in this study, the feasibility of solar-powered UAV delivery work needed to be discussed. Since the subject of this paper was the path planning algorithms proposed for solar-powered UAV delivery systems, we did not expand on how to implement solar-powered UAVs for delivery, we just showed the prototype and proof that it is feasible in modern techniques.

As we discussed before, we chose a transformable solar-powered UAV [22] with solar panels because of the following reasons: (1) it is not possible to build runways for fixed-wing UAVs to take off or land on in crowded cities [24] and (2) it is not practical to use solar-powered UAVs with the solar panel expanded [25] to perform delivery works, since the large area taken over by solar panels must highly reduce the flying speed and safety.

However, the transformable solar-powered UAV from [22] was not practical, since it was not proven to be safe if such types of solar-powered UAVs with payloads were processing a transformation in the sky. In addition, it is difficult for a transformable solar-powered UAV in the fixed-wing state to adjust its flying direction. As a result, this kind of UAV is not quite practical for UAV delivery missions, since the purpose of our solar-powered UAV system was to deliver accurately and on time. Hence, we preferred a new type of transformable solar-powered UAV that only harvested solar energy during the landing time with the solar panel expanded. Additionally, when lifting-off and flying in the sky, the solar panel should be folded to guarantee the speed and safety of the flight. This new prototype of solar-powered UAVs was called transformable delivery solar-powered UAVs (TDSPUs). As a result, the energy consuming model of TDSPUs was like that of normal quadcopters [26], and the energy harvesting model was like that of the solar-powered quadcopter from [27].

At the beginning of the feasibility study of TDSPUs, we needed to make sure solar-powered UAVs could take payloads. From [27], we knew that 88 pieces of SunPower C60 solar cells, each with dimensions of $12.5 \times 12.5$ cm$^2$, required an area of 13.75 m$^2$ and provided 293.92 W of power in standard test conditions. Such solar panels required a payload of 1.2 kg and an extra 1.13 kg due to the construction constraints.

When looking at the most advanced quadcopter designed to carry payloads this year [28], the total mass of the quadcopter was 10.3 kg, and the payload was 5 kg. With the solar panels installed, the additional payload of this quadcopter was still more than 2.5 kg. Despite innovations in solar panel efficiency in recent years, it would be feasible for TDSPUs to take payloads with a maximum mass of 2.5 kg.

Secondly, we needed to find a proper flight duration for TDSPUs, so that we could perform experiments to test our proposed algorithms. The flight duration of the quadcopter needed to be reduced due to the change of structure, and the full payload it took. Thus, let us assume it was reduced from 55 min to 25 min.

Then, based on [28], the advanced quadcopter used a lithium polymer battery, which has a capacity of $2 \times 21,000$ mAh $= 42,000$ mAh. Assume that the battery charging efficiency is 90%, the MPPT efficiency is also 90% (in [29], the battery charging efficiency and MPPT efficiency for lithium polymer batteries charged by SunPower C60 solar cells were both 90%, whereas the other two efficiency parameters in [29] were only used for fixed-wing types of UAVs, so we did not consider them here), so the power harvested by the battery from the 88 pieces of SunPower C60 solar cells can be calculated as $293.92 \times 90\% \times 90\% = 238.08$ W and the battery voltage is 22.2 V. Then, the total energy that could be generated from this battery was $42 \times 22.2 \times 3600 = 3,356,640$ J. Then, the estimated charging time to charge the UAV from 0 to a full battery state was $T_{charge(max)}^{\rho_{max}} = 3,356,640 \div 238.08 = 234.99$ min.

Finally, to improve the safety of the delivery mission, we assumed that the UAVs had a speed of 1m/s [4], so that $d_{max} = 1 \times 25 \times 60 = 1.5$ km. Then, we showed an instance of how a UAV finished a delivery mission during the daytime.

Assume our SPU system could assign a UAV with a mission from node one to node four, and the UAV is able to harvest solar power on node two and node three. The distance between these nodes was: $d_{1 \to 2} = 1; d_{2 \to 3} = 0.6; d_{3 \to 4} = 1.2$. The UAV started its mission at 6:00 am and charged itself on landing places as much as the system assigned.

At time $t_{10} = t_0 + \frac{1}{1.5} \times 25 = 16.67$, the UAV would arrive at node two. Then, assume the peak charging efficiency on node two was 0.6, and the system assigned the UAV to charge on node two to obtain only enough energy to reach the next node. Using Equation (12),

$$t_{11} = \frac{720}{\pi} \sin^{-1} \left( \frac{\pi}{720 \times 0.6} \left( 234.99 \times \frac{0.1}{1.5} + 0.6 \times \frac{720}{\pi} \sin \frac{\pi(16.67 - 360)}{720} \right) \right) + 360 = 111.76. \text{ Then,}$$

with Equation (13), $t_{20} = t_{11} + \frac{0.6}{1.5} \times 25 = 121.76$. Assume the peak charging efficiency in node three was 0.9, still using Equation (12), $t_{21} = \frac{720}{\pi} \sin^{-1} \left( \frac{\pi}{720 \times 0.9} \left( 234.99 \times \frac{1.2}{1.5} + 0.9 \times \frac{720}{\pi} \sin \frac{\pi(121.76 - 360)}{720} \right) \right) + 360 = 371.29$. Then, finally, the time the UAV arrives at the destination node four is $t_{30} = t_{21} + \frac{1.2}{1.5} \times 25 = 391.29$. In other words, the UAV could finish its mission at 12:31 pm.

As a result, TDSPUs were feasible under today's techniques to finish delivery missions. Additionally, we believe that with the development of UAV technology and solar panel technology, such kinds of solar-powered UAVs are likely to become more and more utilitarian in our daily lives.

## 8. Experimental Results

The experiments in this section were performed on MATLAB. We simulated the process of UAV delivery missions on a local city map built with landing places and stores. For each delivery mission, we recorded the total time cost for any single delivery mission using our proposed algorithms and other existing algorithms. The parameters were chosen from Section 7.

*8.1. Simulation of Proposed Algorithms (Statically Charging Efficiency)*

As discussed in Section 2, previous studies about UAV delivery path planning problems could be classified as Dijkstra-based algorithms and heuristic algorithms. Since it is difficult to determine the parameters in previously proposed heuristic algorithms for SPU path planning problems in a dynamically charging efficiency environment, we only compared our proposed algorithms with Dijkstra-based UAV path planning algorithms.

The GOA was compared with the following two kinds of Dijkstra-based UAV path planning algorithms according to the total mission time:

1. Distance-based Dijkstra algorithm which did not apply our optimal CTA algorithm (D-Dij)

(Distance-based Dijkstra means the cost function in the Dijkstra algorithm only concerns flying time, which was mentioned in [8]. A Dijkstra algorithm that did not apply the optimal CTA algorithm meant that the UAVs always charged themselves to a full battery state. In addition, to make the comparison fairer, if the UAV arrived at a node that could reach the destination on path P, the UAV only charged enough power to reach the destination instead of to a full battery state. This operation exhausted the UAV's battery when it arrived at the destination, just like what we assumed in Section 4).

2. Weight-based Dijkstra algorithm which did not apply the optimal CTA algorithm (W-Dij)

(Weight-based Dijkstra means the cost function in the Dijkstra algorithm was calculated through the summation of the flying time to reach the next node and the charging time on the next node to compensate for the energy consumed during this flight [8].)

Each time running a case, we generated an H × H km$^2$ local city map. The density of landing places varied from 40 to 60 and 80 per 49 km$^2$, with a uniform distribution.

Assuming that $T_{total(GOA)}$ represents the averaged total mission time for the pruned GOA (Algorithm 2), and $T_{total(Dij)}$ represents the Dijkstra algorithm applied in [2], the percentage of time saved by the GOA algorithm was:

$$Ef_{GOA} = \frac{T_{total(Dij)} - T_{total(GOA)}}{T_{total(Dij)}} \tag{28}$$

The following diagrams describe when H was fixed, the minimum charging efficiency (MCE) in the local city map ranged from 0.1 to 0.9 (nine points total) and the percentage of time saved by the GOA algorithm compared with different kinds of Dijkstra algorithms. The improvement was measured using Equation (28), and we only showed the maximum improvement.

Based on Figure 10, we could conclude that our GOA algorithm showed a remarkable improvement compared with the Dijkstra algorithms. As the MCE increased, the improvement by the GOA compared with D-Dij dropped linearly, while the improvement by the GOA compared with W-Dij would not decrease so quickly when the MCE moved closer to one.

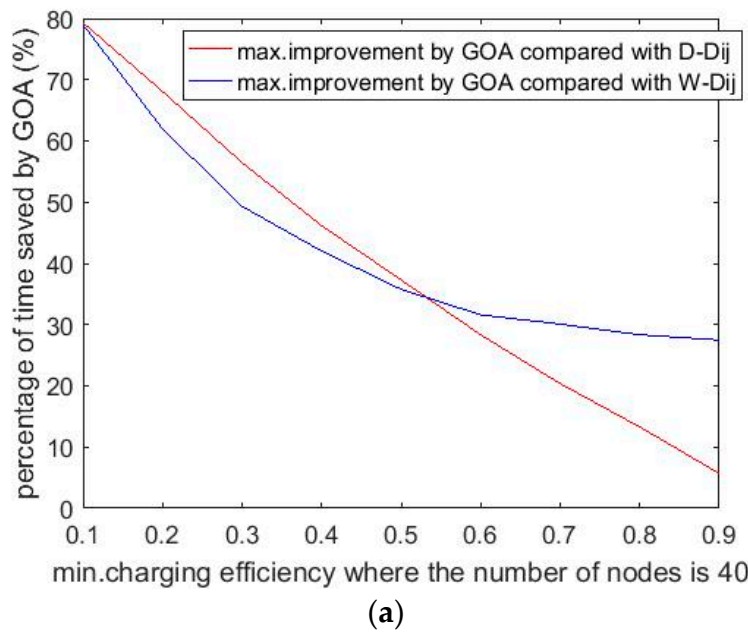

(**a**)

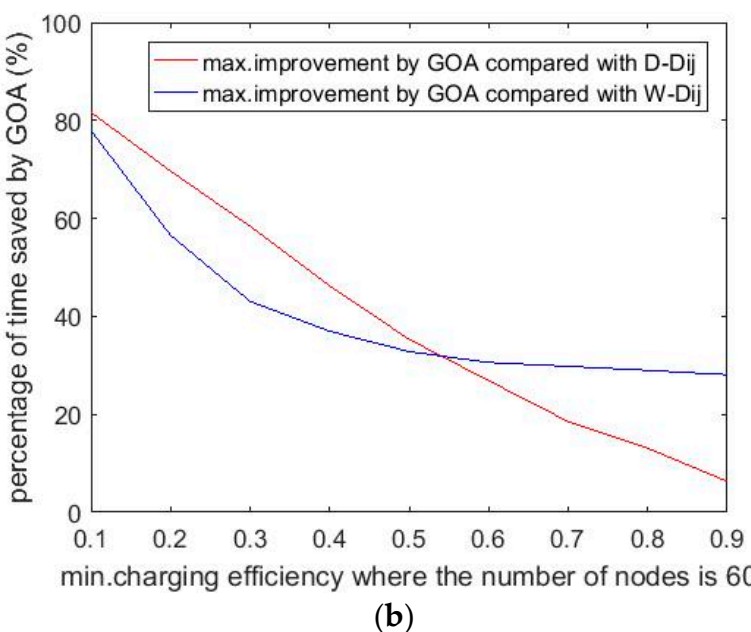

(**b**)

**Figure 10.** *Cont*.

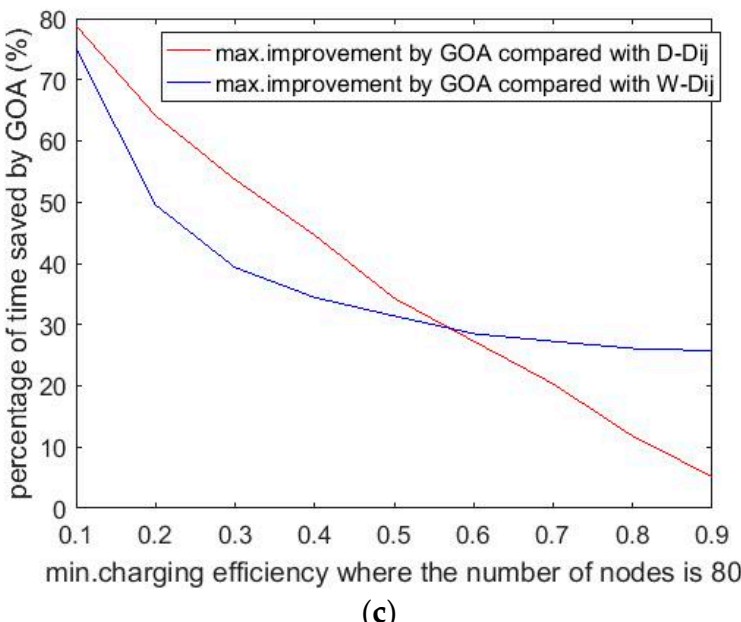

**(c)**

**Figure 10.** (**a–c**) Maximum improvement by the GOA, where the number of nodes were 40, 60, and 80 (in Figure 10, max. improvement represents the maximum improvement an algorithm could achieve, and min. charging efficiency represents the minimum possible charging efficiency in the city map).

These great improvements happened in special cases, where the following situations occurred:

1.  None of the paths found by the Dijkstra algorithm were the optimal path.
2.  The CTA plan by the Dijkstra algorithm was not optimal.
3.  The variance of charging efficiency along the candidate optimal paths was large.

The improvement by the CTA algorithm contributed more to the experiment results for special cases. In some paths, if we used the CTA plan of the Dijkstra algorithms, UAVs would always charge themselves to a full battery state. However, if the charging efficiency of landing places in the early trip was low, and the charging efficiency of landing places in the later trip was high, our CTA algorithm charged the UAV less early in the trip, and then charged more time later in the trip; then, the total charging efficiency for the mission was improved. If the charging efficiencies for nodes on such paths had a large difference, the optimal CTA plan chosen by our CTA algorithm could obtain a large percentage of improvements when compared with the Dijkstra-based CTA plans. The different path planning results between the GOA and Dijkstra algorithms also contributed to the improvement, since the paths found by the Dijkstra algorithms were sometimes not the optimal path (proved in Section 4). As a result, in the situation that nodes in the candidate optimal paths had large differences (or, in other words, the variance of charging efficiency along the candidate optimal paths was large), the GOA could achieve great improvements when compared with the Dijkstra algorithms. For example, in Figure 10a, the GOA (Algorithm 2) could save 80% mission time compared with Dijkstra-based algorithms when the MCE was 0.1 on a local city map. Let $T_{running(pruned)}$ represent the total running time for the GOA with pruning, and $T_{running(unpruned)}$ represent the GOA without pruning; therefore, the ratio of $T_{running(unpruned)}$ to $T_{running(pruned)}$ would be:

$$ER_{pruning} = \frac{T_{running(unpruned)}}{T_{running(pruned)}} \tag{29}$$

Then, the $ER_{pruning}$ using the GOA with pruning compared with the unpruned GOA is shown in the following diagram:

The result from Figure 11 shows that the pruning strategy could exponentially reduce the running time of the GOA when the number of nodes in the local city map increased.

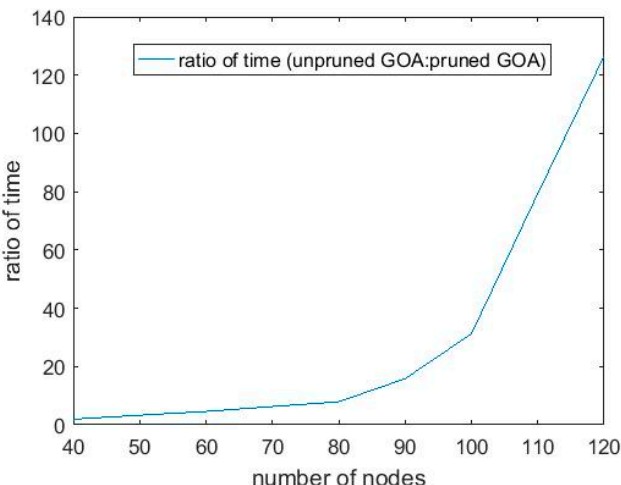

**Figure 11.** Ratio of time (unpruned GOA: pruned GOA).

*8.2. Simulations of Proposed Algorithms (Dynamically Charging Efficiency)*

For the experiment in the dynamically charging efficiency scheme, our proposed DG-CTA algorithm was compared with the CTA plan proposed by the Dijkstra algorithms (always charging the UAV to a full battery state) and the CTA algorithm using Algorithm 1, which we proposed in Section 4. Since this was a comparison between CTA algorithms, the paths needed to be fixed. We chose every ten paths that had the minimized total Dijkstra distance in each topology and applied different CTA algorithms.

The heuristic algorithm combined with the DG-CTA algorithm was compared with W-Dij in this experiment as well.

These special cases from Figure 12 described by the red curve happened when Algorithm 1 had an opposite local CTA plan to Algorithm 3. For example, if the charging efficiency in the first landing place was a little higher than the second node, then Algorithm 1 assigned the UAV to charge to a full battery state. However, since it was early in the morning, the charging efficiency on all the nodes was low. Then, our DG-CTA algorithm assigned the UAV to charge only long enough to have enough energy to reach the next node, because the charging efficiency increased dramatically in the morning, and it was more efficient to charge on the next node. As a result, based on Operation 5.4, in the case of when the UAVs arrived at nodes with higher charging efficiencies in the morning or the late afternoon, the DG-CTA algorithm had different results compared with Algorithm 1; then, the improvement by the DG-CTA was obvious.

Described by the blue curve, the special cases where the DG-CTA had a great improvement compared with the weighted Dijkstra-based CTA plans happened when the DG-CTA had a remarkable improvement compared with Algorithm 1 and the variances of charging efficiency along the paths were large.

The special cases described by the green curve that our heuristic algorithm had a great improvement compared with the weighted Dijkstra algorithm happened when the DG-CTA algorithm already had a great improvement and there was a better path chosen by the heuristic algorithm compared with the Dijkstra algorithms.

In conclusion, although our DG-CTA-based heuristic algorithm was not the globally optimal solution, it could still achieve a large amount of improvement compared with existing CTA algorithms. Moreover, our proposed heuristic algorithm could achieve a great improvement compared to the existing Dijkstra algorithm in dynamically charging efficiency schemes with an acceptable running time. For example, in Figure 12a, Algorithm 4 could save 70% of the mission time compared with the Dijkstra-based algorithms when the MCE was 0.1 on a local city map.

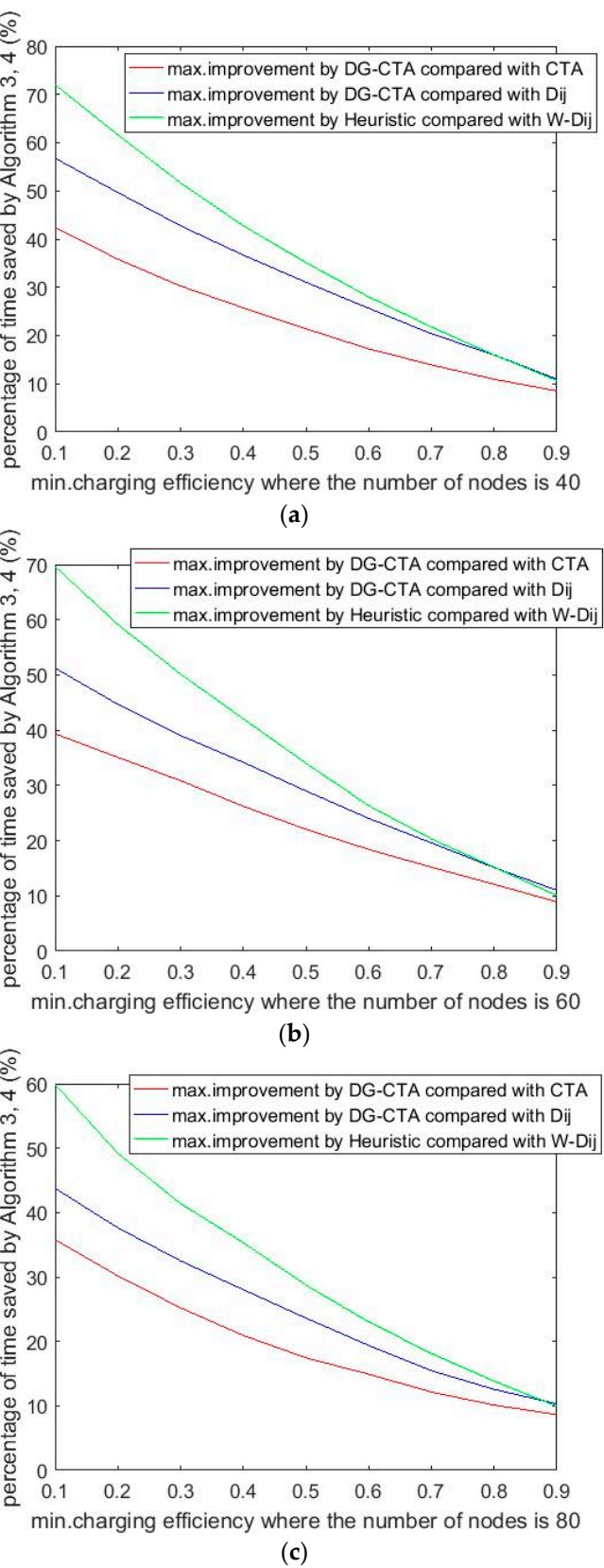

**Figure 12.** (**a**–**c**) Maximum improvements by Algorithms 3 and 4, where the number of nodes were 40, 60, and 80 (in Figure 12, DG-CTA represents Algorithm 3, and heuristic represents Algorithm 4; CTA represents Algorithm 1, and Dij represents the scheme where UAVs always charged themselves to a full battery state on each passed node).

## 9. Conclusions

In this paper, a solar-powered UAV delivery system was proposed to extend the endurance of UAVs for delivery missions. To find the globally optimal solution to the UAV path planning problem in the statically charging efficiency scheme, we proposed a very efficient pruning-based globally optimal algorithm. When considering the routing solution according to the dynamically charging efficiency environment, we also proposed a DG-CTA-based heuristic algorithm. Finally, we designed mission arrangement protocols to manage the UAV delivery missions and solve system-level issues in the SPU delivery system. Simulation results showed that our proposed algorithms had significant improvements compared with previous algorithms when applied in the SPU schemes.

For future research, it is suggested that solar-powered UAVs should be able to automatically find the landing places during voyages by themselves. In addition, we plan to study the path planning problem and the system model in UAV delivery systems with more dense landing places deployed.

**Author Contributions:** Conceptualization, Z.J.H. and Z.T.; methodology, Z.T.; software, Z.T.; validation, Z.J.H., Z.T. and S.S.; formal analysis, Z.T.; investigation, Z.T.; resources, Z.J.H.; data curation, Z.T.; writing—original draft preparation, Z.T. and S.S.; writing—review and editing, Z.T., Z.J.H. and S.S.; visualization, Z.T.; supervision, Z.J.H.; project administration, Z.J.H. and Z.T.; funding acquisition, Z.J.H. All authors have read and agreed to the published version of the manuscript.

**Funding:** This research was funded in part by the U.S. National Science Foundation grant numbers CNS-1763627 and DGE-1820640.

**Institutional Review Board Statement:** Not applicable.

**Informed Consent Statement:** Not applicable.

**Data Availability Statement:** Data sharing not applicable.

**Conflicts of Interest:** The authors declare no conflict of interest.

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
