# Peer review of "Routing in Solar-Powered UAV Delivery System"

_drones, doi:10.3390/drones6100282_

Round 1

Reviewer 1 Report (Previous Reviewer 2)

The authors have responded carefully to the  comments with additional simulation results added. All the questions and concerns have been answered and addressed appropriately. I therefore would recommend to accept the manuscript for publication.

Author Response

We would like to thank you for reviewing our paper and for your constructive comments and suggestions. We have addressed all your comments. Below, we summarize, comment-by-comment, how we have revised our manuscript in response to your raised concerns. We believe the manuscript is now in excellent shape for publication, and we kindly request your approval.

Reviewer 2 Report (New Reviewer)

The authors proposed a solar-powered UAV delivery system and illustrated the results in both statical and dynamically charging efficiency environments. Firstly, English writing must be improved. It is hard to read because of writing style and grammatical mistakes. I strongly suggest the authors, hire an English proficient editor would greatly improves the language of the paper. Other major comments are:

1. besides English writing, the authors repeatedly use future tense like the use of a word 'will'. Although this is current work. Among the common mistakes, dynamical should be corrected as dynamically. there are so many such common mistakes.

2. Lacks a clear description of the novelty and contribution of the proposed algorithm.

3. line 95: define NP

4. line 126: authors assumed that UAVs charge themselves to full battery state without CTA plan. 

5. line 129: which modifies the linear ... (avoid using future tense)

6. line 130: vehicle recharging in the UAV delivery system.

7. line 138: path planning algorithms show ...

8: line 182: SPU problem is designed to find ...

9: Figure 2: Is node 6 also the reachable node as stated in the text? but node 6 lies outside the circle in Figure 2.

10: line 299: where m_c denotes  ____?

11: line 300: where UAV_b denotes _____?

12: Figure 4: are these store nodes represent the charging stations?

13: headings of sections 3.2 and 4 looks similar and maybe other section as well. 

14: Define DNCS.

15: line 323: path found by ...

16: line 329: we need a principle ...

17: line 565: R_max could be _______? (what is value of R_max ?)

18: line 999: Always write an equation immediately after it was cited. Eq. (30) is missing

19: Figure 10: I think, there are some mistakes in the legend label, which one curve shows the GOA curve? figures are not clear but blurred. Replace figures 10 to 12 by high-quality figures.

20: In figure11: is the vertical axis the total running time or a ratio? because the ratio is unitless. if it is the time axis then it seems like an increasing number of nodes exponentially increases the running time of GOA.

Also, cite Figure 11 in the text.

21: The legend label in Figure 12 should be written precisely. it's really confusing. It should be written as; max. improvement by DG-CTA, max. improvement by Dijkstra, max. improvement by heuristic DG-CTA algorithm. or otherwise; are these curves show a comparison of two methods? is the green curve show a comparison of two different methods? if yes, then which one is DG-CTA and Algorithm 1? and so on...

Cite Figure 12 in the text immediately after or before the figure.

22: Please state the software platform where the experiments were performed. And how the data was collected for path planning in the delivery system. 

23: there are many mathematical equations written within text paragraph. is it possible to separate them in tabular form or some other good way?

Thank you for your hard work and contribution to the journal. 

good luck 

Author Response

We would like to thank you for reviewing our paper and for your constructive comments and suggestions. We have addressed all your comments. Below, we summarize, comment-by-comment, how we have revised our manuscript in response to your raised concerns. We believe the manuscript is now in excellent shape for publication, and we kindly request your approval.

Round 2

Reviewer 2 Report (New Reviewer)

The authors have addressed all the necessary comments in the revised version of the manuscript. Some minor corrections:

1. line 137: without CTA plan consideration

2. (where m-c denotes Cth mission)

3. (where UAV_B denotes Bth UAV)

This manuscript is a resubmission of an earlier submission. The following is a list of the peer review reports and author responses from that submission.

Round 1

Reviewer 1 Report

Summary of the paper and Strength of it:

The authors proposed an interesting system for delivering goods in the urban area using UAVs. They are powered using solar energy instead of using dedicated charging stations that may cost a lot of money. Hence, the goal is ambitious by supporting the cities by a cost-effective solution.  

Weaknesses:

-         The main concern of the paper is the practicality of the proposed system. Although charging a micro-drone using solar energy may be practical when the required traveled distance is not large and the application is not delay sensitive, powering UAVs that are used for carrying goods using solar energy may not be practical. The authors did not provide a practical scenario with complete specifications for a UAV carrying a certain payload, and accurately and correctly calculating the time of charging, flight time, and the mission completion time. When these times are calculated accurately, we can conclude the practicality of adopting such systems using current state-of-the arts technologies.

-         The assumptions are not consistent, and that may make the results not accurate, e.g.,: in line 751-756 of the paper, it is mentioned that the used capacity of the battery in our study is 9500 mAh based on reference [16]. However, reference [16] mentioned that the capacities of used drones are 6600 mAh and 23800 mAh. Then, it is mentioned that the used drone is the transformable drone in [14] with battery SunPower C60. However, reference [14] mentioned that the battery is SunPower E60 with capacity 3600 mAh. Hence, which battery is really assumed to be used? With capacity 9500 mAh, 6600 mAh, 23800 mAh or 3600 mAh? Then, an efficiency of 40% is assumed based on reference [18]. However, the charging time should be calculated based on the efficiency and the other specification of the transformable UAV mentioned in reference [14] to be consistent. In summary, the charging time must be accurate, and the assumptions should be practical and based on a certain drone. Moreover, the used drone is not designed for carrying goods. Hence, different technical specifications for the drone including a larger battery and much longer charging time is expected.

-         Although the proposed system is proposed for delivery, the paper does not consider the weight of the payload. The weight of the payload that can be carried and the total weight of the drone is very important in deciding the flight time and the maximum traveled distance given a certain battery capacity and drone specifications.

-         It is not clear why the Assumption 3.2.1 is assumed, where the optimization problem should minimize the total flight time, which means that the UAV routing should not allow the UAV to go back on its route.

-         The authors derived the local optimal solution because the original problem is NP-hard. However, it is not clear how good the proposed solution is compared with the optimal solution. Although finding an optimal solution for NP-hard may not be possible for the large problem, some optimization tools can derive the optimal solution when the scale of the problem is very small. Deriving the optimal solution for a small-scale problem (e.g., small number of nodes) or using the brute force method for a small-scale size then comparing the optimal solution with the proposed solution is appealing.

-         The algorithms in the paper should be described in detail with reference to the line number in the algorithm. The purpose of each line or group of lines should be described in addition to describing the whole purpose of the algorithm.

-         Related works sections need to discuss more related works. It is mentioned that few works related to solar-powered UAV routing are in the literature. However, only one reference related to solar-power UAV routing is discussed. More related works to solar-powered UAV charging and path planning are needed. A quick search about related works, we can find some works related to solar-powered UAV routing not included in this paper. Related works should be comprehensive, organized and present the similarity and difference between the proposed work and the related works.

Reviewer 2 Report

This paper discusses an interesting and generally useful topic in the field of unmanned aerial vehicles, namely the path planning problem for solar-powered unmanned aerial vehicle cargo distribution systems, which represents a challenging robotics subfield. A path node selection strategy is derived and simulated numerically. The article is well organized and the results presented seem to be correct. However, the revision should have taken into account the following issues:

1) A literature review of the system family of solar-powered delivery system path planning in this paper should yield a better structure. It should lead to an introduction to what the motivation for this article is, what has not been done before, and why the authors are dealing with and researching this issue.

2) The authors should establish an environmental model of the drone city distribution system for practical problems, as well as a model of the energy input and output of the drone.

3) The position of formula (1) and its argument interpretation do not match the context logic, it should be placed in an appropriate place.

4) It is very good to derive the proposed algorithm in detail, but its better to explain how the actual application of the proposed algorithm and the testing process are carried out.

5) Algorithms 1 and 2 give degrees of time freedom, which is rigorous, but unfortunately algorithms 3 and 4 are not given. For the convenience of the reader, the author should briefly give the computational process of the computational complexity of all proposed algorithms in order to design an optimization strategy.

6) The example used to demonstrate the main results needs to be improved. More discussions should be given to clearly demonstrate the effectiveness of the obtained results. Does high charging efficiency mean that less time is spent completing cargo distribution tasks? The author should provide directly relevant results to illustrate that the proposed algorithm can effectively reduce the time to distribute goods.

7) The results given by the authors of the seventh chapter are all improvements compared to Dijkstra algorithms. The authors should independently compare Dijkstra algorithms with the proposed algorithms in solving the same problem to save time. Where possible, it is recommended that the author compare it with other classical algorithms of the shortest path of graphs (e.g., Bellman-Ford algorithm) to verify the superiority of the proposed algorithm.

8) Why the speed of the drone is set to constant, we know that when the drone may be in flight because of some unexpected factors to make the drone do variable speed movement; In addition, what if the distance between the two nodes is greater than the range of the drone? When the drone transports goods of different weights, it will also affect the endurance time of the drone, and the above factors will affect the completion time the author should analyze.

9) Is the drone landing site planned in advance, or is it temporary? For example, on the roof of a building, whether planned or not, when the UAV performs its mission, it finds that the landing point is safe and the charging time can be reduced. Will it land at this point?

10) The author should consider the generalization of the proposed algorithm. When migrating a model to a new environment, is it going to work just as well?

11) Parallel picture width is inconsistent, should use the original picture to avoid screenshots, pay attention to the appearance. The special name case in the caption should be consistent with the text. Multiple result pictures, each with a numbered description (a, b, c....). Note the time abbreviations in the axes in the figure. The problem of quoting pictures in the paper.

12) Formulas and numbers should be edited as required by the journal. The formula in the algorithm pseudocode is mixed with positive italics and should be modified consistently. Some parts of the paper are inconsistent in font format and size.

13) Attention should be paid to the indentation of algorithm pseudocode, each line of code should be numbered (1, 2, 3......).

14) References and citations in articles should be edited according to journal requirements.

15) The logic of the article needs to be sorted out to reduce the repetition of variable definitions in the article.